# Observations and modeling of areal surface albedo and surface types in the Arctic

Evelyn Jäkel[1], Sebastian Becker[1], Tim R. Sperzel[1], Hannah Niehaus[2], Gunnar Spreen[2], Ran Tao[3], Marcel Nicolaus[3], Wolfgang Dorn[4], Annette Rinke[4], Jörg Brauchle[5], and Manfred Wendisch[1]

[1]Leipzig Institute for Meteorology, Leipzig University, Germany
[2]Institute of Environmental Physics, University of Bremen, Bremen, Germany
[3]Alfred Wegener Institute, Helmholtz Centre for Polar and Marine Research, Bremerhaven, Germany
[4]Alfred Wegener Institute, Helmholtz Centre for Polar and Marine Research, Potsdam, Germany
[5]Institute of Optical Sensor Systems, German Aerospace Center, Berlin, Germany

**Correspondence:** Evelyn Jäkel (e.jaekel@uni-leipzig.de)

**Abstract.** An accurate representation of the annual evolution of surface albedo of the Arctic Ocean, especially during the melting period, is crucial to obtain reliable climate model predictions in the Arctic. Therefore, the output of the surface albedo scheme of a coupled regional climate model (HIRHAM–NAOSIM) was evaluated against airborne and ground-based measurements. The observations were conducted during five aircraft campaigns in the European Arctic at different times of the year between 2017 and 2022; one of them was part of the Multidisciplinary drifting Observatory for the Study of Arctic Climate (MOSAiC) expedition in 2020. We applied two approaches for the evaluation, (a) relying on measured input parameters of surface type fraction and surface skin temperature (offline), or (b) using HIRHAM-NAOSIM simulations independently of observational data (online). From the offline method we found a seasonal-dependent bias between measured and modeled surface albedo. In spring, the cloud effect on surface broadband albedo was overestimated by the surface albedo parametrization (mean albedo bias of 0.06), while the surface albedo scheme for cloudless cases reproduced the measured surface albedo distributions for all seasons. The online evaluation revealed an overestimation of the modeled surface albedo resulting from an overestimation of the modeled cloud cover. Furthermore, it was shown that the surface type parametrization contributes significantly to the bias in albedo, especially in summer (after the drainage of melt ponds) and autumn (onset of refreezing). The lack of an adequate model representation of the surface scattering layer, which usually forms on bare ice in summer, contributed to the underestimation of surface albedo during that period. The difference of modeled and measured net irradiances for selected flights during the five airborne campaigns was derived to estimate the impact of the model bias for the solar radiative energy budget at the surface. We revealed a negative bias between modeled and measured net irradiances (median: -6.4 W m$^{-2}$) for optically thin clouds, while the median value of only 0.1 W m$^{-2}$ was determined for optically thicker clouds.

## 1 Introduction

The decline of sea ice and snow cover of the Arctic Ocean due to a warming climate leads to a decrease of the surface reflection (albedo) and, therefore, causes an increase of absorption of solar radiation incident at the ocean surface, which enhances

surface temperature and sea ice cover decline even further. This positive surface albedo feedback is one of the major drivers of Arctic amplification (Screen and Simmonds, 2012; Pithan and Mauritsen, 2014; Goosse et al., 2018), which comprises a higher than globally averaged warming of the Arctic near-surface air temperature (Serreze et al., 2009). Although the surface

albedo feedback is generally understood, its impact on Arctic amplification is hard to quantify (Qu and Hall, 2014; Block et al., 2020; Taylor et al., 2022). This feedback has direct implications in the sunlit season, whereas it contributes indirectly to Arctic amplification in autumn and winter (Dai, 2021; Wendisch et al., 2019). Furthermore, as a consequence of thermodynamic forcing and changes of ice dynamics, a general shift from older, thicker sea ice to younger, thinner ice is observed (Kwok, 2018; Li et al., 2022) affecting the heat storage of the Arctic Ocean mixed layer (Arndt and Nicolaus, 2014; Stroeve et al., 2014;

Perovich et al., 2020) and the winter energy balance.

The exchange of radiative energy fluxes at the atmosphere-ocean interface in summer particularly depends on the timing of the melt onset and the progress of melting. This period is poorly projected in climate models (Mortin et al., 2014). Also the consequences of the melting onset influences evolution of surface albedo, which appears crucial to obtain reliable estimates from climate models (Liu et al., 2007; Wyser et al., 2008; Toyoda et al., 2020). In models, various sea ice albedo parametriza-

tions with different complexity are applied (Pirazzini, 2009; Thackeray et al., 2018). As the spatial scale of the surface type variation is smaller than common grid sizes of climate models, the surface albedo schemes commonly include a parametrization of the fractions of different surface types (melt ponds, bare ice, snow). The parametrization of the albedo of the respective sea ice surface types is usually based on a temperature-dependent function describing the transition between dry and wet surface conditions. More complex parametrizations account for snow aging as a function of time since the last snowfall (Wyser

et al., 2008). Liu et al. (2007) have shown that other surface albedo parametrizations using additional parameters, such as snow depth and spectral band dependence, can yield more realistic regional variations of ice distributions. Moreover, Pedersen and Winther (2005) identified the driving meteorological parameters (temperature, snow depth, days with temperature above 0 °C) for modeling the snow albedo by applying a multi-linear regression based on field measurements. Furthermore, observations in combination with radiative transfer simulations have proven a relevant effect of spectral cloud absorption on the snow broad-

band albedo (e.g., Grenfell et al., 1994; Gardner and Sharp, 2010; Pirazzini et al., 2015; Jäkel et al., 2019). This dependence is included in only few surface albedo schemes (Jäkel et al., 2019; Boucher et al., 2020; van Dalum et al., 2020).

Evaluations and adjustments of surface albedo parametrizations are usually based on field observations (Curry et al., 2001; Køltzow, 2007; Liu et al., 2007; Jäkel et al., 2019; Toyoda et al., 2020; Light et al., 2022), preferably covering the annual course of surface properties, as provided, for example, by the Surface Heat Budget of the Arctic Ocean project (SHEBA, Pers-

son et al., 2002) or the Multidisciplinary drifting Observatory for the Study of Arctic Climate (MOSAiC) expedition (Light et al., 2022; Nicolaus et al., 2022). However, evaluating models based on local-scale observations is difficult because single-point measurements may not be representative of the large grids used in climate models, especially during the melt season. On the other hand, validations against satellite observations (e.g., Qu and Hall, 2014; Thackeray et al., 2018) are restricted to cloudless situations, which limits the temporal resolution of satellite-based surface albedo measurements. As a compromise,

airborne observations provide data covering different atmospheric conditions on a larger spatial scale, partly resolving the sub-grid variability in a model grid cell. However, these observations are limited in time to a few weeks per year.

Nevertheless, we have compiled observational data of sea ice albedo from five airborne campaigns, covering spring (March/April), summer (May/June), and autumn (September) conditions. We use this data set to evaluate the surface albedo scheme of the coupled regional climate model HIRHAM–NAOSIM (Dorn et al., 2019). This scheme was recently updated with a cloud-cover-dependent snow albedo parametrization and an adjustment of temperature thresholds based on airborne surface broad-band albedo measurements performed in the north of Svalbard in early summer 2017 (Jäkel et al., 2019). A comparison of the modeled surface albedo between the revised model and the earlier version was presented by Foth et al. (2023). They evaluated both model versions using measurements from two flux stations that were deployed during MOSAiC. They found that the revised snow surface albedo parametrization led to a more realistic simulation of surface albedo variability during the snow melt period in late May and June.

In this study, the accuracy of the revised surface albedo scheme for different seasons and regions is quantified. First, the parametrizations were run offline by using input parameters from our airborne measurements (offline evaluation). Second, the HIRHAM-NAOSIM model was operated independently (online evaluation) for different time periods, and the output was directly compared with observations. For this purpose, Section 2 presents the measured data set and parametrization scheme of HIRHAM-NAOSIM along with the model-measurement comparison methodology. The spatiotemporal variability of the measured surface types and surface albedo are discussed in Section 3. The evaluation results for different applications are presented and seasonal differences in the accuracy of the model results are examined. Section 4 quantifies the impact of a possible bias in surface albedo on the solar radiative energy budget in terms of differences in net irradiance.

## 2 Materials and methods

### 2.1 Study area and campaigns

Five flight campaigns were conducted within the collaborative research project "Arctic Amplification: Climate Relevant Atmospheric and Surface Processes, and Feedback Mechanisms" $(\mathcal{AC})^3$ between 2017 and 2022 (Wendisch et al., 2023). The airborne activities were carried out with the research aircraft Polar 5 (P5), Polar 6 (P6) (Wesche et al., 2016) and focused on observations of Arctic clouds, of the Arctic atmospheric boundary layer, and surface properties in the European Arctic in different seasons between 2017 and 2022. Table 1 gives an overview of the campaigns considered in this study. Apart from the "Polar Airborne Measurements and Arctic Regional Climate Model Simulation Project" (PAMARCMiP) campaign, which was based at the Danish Villum research station (Station Nord), Greenland (81° 36' N, 16° 40' W), all campaigns were based in Longyearbyen (78° 13' N, 15° 38' E), Svalbard. The "Arctic CLoud Observations Using airborne measurements during polar Day" (ACLOUD) took place in early summer 2017 and used both, P5 and P6. Spring Arctic conditions were observed with P5 only during PAMARCMiP and during the "Airborne measurements of radiative and turbulent FLUXes of energy and momentum in the Arctic boundary layer" (AFLUX) campaign. Within the most recent spring observations in 2022, HALO-$(\mathcal{AC})^3$, both aircraft, P5 and P6, were used with a similar instrumental setup as during ACLOUD. The set of airborne observations during polar day was completed by the "MOSAiC Airborne observations in the Central Arctic" (MOSAiC-ACA) campaign, which took place in autumn 2020, as an airborne component of the MOSAiC expedition.

**Table 1.** List of campaigns and time frame of observations. Number of used data points is given in total for Polar 5 (P5) and Polar 6 (P6) flights. SZA stands for solar zenith angle.

| Campaign | Season | SZA (°) | Number of data points | Number of research flights | Campaign reference |
|---|---|---|---|---|---|
| ACLOUD | May/June 2017 | 56 - 69 | 4061 | 21 (P5 + P6) | Wendisch et al. (2019); Ehrlich et al. (2019) |
| PAMARCMiP | March/April 2018 | 75 - 84 | 5545 | 7 (P5) | Jäkel et al. (2021) |
| AFLUX | March/April 2019 | 73 - 82 | 527 | 6 (P5) | Mech et al. (2022) |
| MOSAiC-ACA | September 2020 | 71 - 78 | 11079 | 5 (P5 + P6) | Mech et al. (2022) |
| HALO-$(\mathcal{AC})^3$ | March/April 2022 | 73 - 83 | 802 | 4 (P5) | Wendisch et al. (2021) |

For this study, only low-level (flight altitude below 300 m) flight sections were considered, which were performed above sea ice and open water without clouds between the aircraft and surface. This selection was made to minimize atmospheric masking effects in the albedo observations (Wendisch et al., 2004). Further, the data were filtered with respect to aircraft pitch and roll angles within a range of $\pm 4°$ to reduce the uncertainties of radiation measurements due to horizontal misalignment (Wendisch et al., 2001). Figure 1 shows the coverage of all selected flight sections for the five aircraft campaigns, together with the sea ice edge as defined from the satellite observation of the sea ice concentration (SIC) derived from the Advanced Microwave Scanning Radiometer2 (AMSR2) instrument using the method from Spreen et al. (2008). Here, the ice edge is based on 80 % SIC. The strong variations of the sea ice edge position is not only linked to the period of observations, but also with interannual changes. While the northernmost retreat of sea ice was observed during MOSAiC-ACA in autumn, the southern sea ice edge in the measurement area during ACLOUD in early summer and during the two spring campaigns HALO-$(\mathcal{AC})^3$ and AFLUX does not differ so much. A relatively far northern location of the ice edge for spring conditions was observed during PAMAR-CMiP in 2018. However, due to the more northern starting point of the aircraft in Station Nord, Greenland, mostly surfaces with more than 80 % of SIC could be overflown. For MOSAiC-ACA, a significant fraction of measurements were carried out in the marginal sea ice zone (MIZ), while for the other campaigns mainly flight sections with SIC larger than 80 % remained after application of the selection criteria. The total number of data points and flight days are listed in Table 1. Most flights over sea ice were performed during ACLOUD. However, the largest number of selected measurements was determined for MOSAiC-ACA, where the instrumental setup of the P6 differed from the configuration of the P5, resulting in different time resolutions of the data products.

## 2.2 Instrumentation and products

### 2.2.1 Radiation measurements

Broadband irradiance measurements (200–3600 nm wavelength, referred to as solar in the following text) were performed by a pair of pyranometers (CMP22 by Kipp&Zonen, Delft, The Netherlands) installed on top and bottom of the aircraft fuselage on both AWI aircraft. The manufacturer gives an irradiance measurement uncertainty of about 2 %. This value increases for a

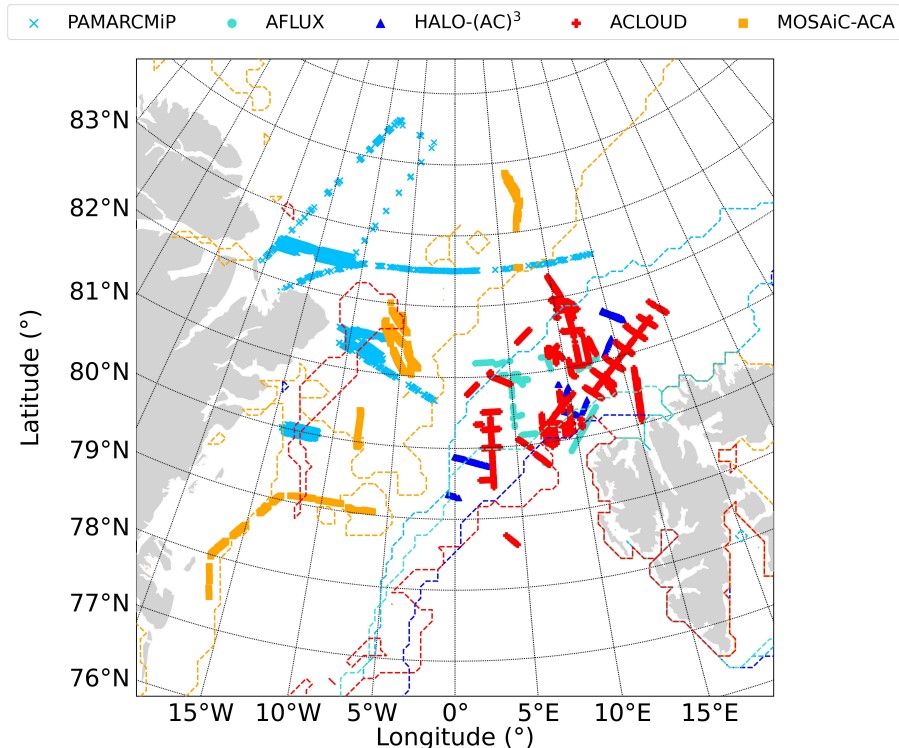

**Figure 1.** Flight sections of surface albedo measurements for all campaigns listed in Tab.1. The sea ice edge (based on 80 % SIC) representative for each campaign is plotted with the same color code as the flight track but with dashed lines.

higher solar zenith angle (SZA) due to the increase of the cosine response error (maximum $\pm 3\%$ deviation from ideal at 80°
SZA). The irradiance data were corrected for aircraft pitch and roll attitude angles following the method described by Bannehr
and Schwiesow (1993). A deconvolution technique was applied to the pyranometer measurements to enhance the temporal
resolution (20 Hz) of the slow-response sensors, as proposed by Ehrlich and Wendisch (2015). The areal surface broadband
albedo along the selected flight sections was derived from the ratio of upward and downward solar irradiances.

The spectral surface albedo was derived from the spectral modular airborne radiation measurement system (SMART) installed
on board of the P5 during all campaigns apart from AFLUX (Wendisch et al., 2001). The optical inlets mounted on top and
bottom of the aircraft fuselage were actively stabilized to correct for aircraft movement. A set of four spectrometers (two for
each hemisphere) covered a spectral range from 300 nm to 2200 nm wavelength with a resolution of 3 nm (below 900 nm
wavelength) and 9 – 15 nm (from 900 nm wavelength) (Bierwirth et al., 2009; Jäkel et al., 2013). The measurement uncertainty
of the downward and upward spectral irradiances was estimated at 5.7 % and 4.0 %, respectively (Jäkel et al., 2021).

During the MOSAiC expedition, autonomous radiometers were installed on the sea ice. Each radiation station consisted of
three RAMSES-ACC-VIS radiometers manufactured by TriOS 105 GmbH, Rastede, Germany, measuring spectral upward and
downward spectral irradiances from 320 nm to 950 nm with a spectral resolution of 3.3 nm (Nicolaus et al., 2010). The third

sensor is used to derive the transmitted spectral irradiance through the sea ice. Nicolaus et al. (2010) estimated an uncertainty of less than 5 % for all wavelengths and zenith angles. Potential sensor tilt effects were monitored by comparing radiative transfer simulations, assuming cloudless conditions, with the diurnal pattern of the measured downward spectral irradiance.

Under overcast conditions, misalignment effects were considered to be of minor importance. Further, all data measured for a SZA higher than 85° were excluded.

Since the RAMSES-ACC-VIS radiometers do not cover the entire solar spectral range, an empirical correction function was applied to convert the measured surface spectral albedo into the surface broadband albedo covering the entire solar spectral range. This correction was derived from collocated broadband albedo measurements taken by pyranometers and measurements

of the spectral upward and downward irradiances ($F_\lambda^\uparrow$, $F_\lambda^\downarrow$) from the SMART instrument during ACLOUD. The SMART irradiances were spectrally integrated between $\lambda_1 = 320\,\mathrm{nm}$ and $\lambda_2 = 950\,\mathrm{nm}$, corresponding to the range of the RAMSES-ACC-VIS radiometers. Then the derived integrated surface albedo was separately correlated with the broadband pyranometer data for cloudless and cloudy conditions, yielding two correction functions that account for the missing spectral range:

$$\alpha_{\mathrm{bb}} = 0.779 \cdot \frac{\int_{\lambda_1}^{\lambda_2} F_\lambda^\uparrow \, \mathrm{d}\lambda}{\int_{\lambda_1}^{\lambda_2} F_\lambda^\downarrow \, \mathrm{d}\lambda} + 0.074 \qquad \text{(cloudless) and} \tag{1}$$


$$\alpha_{\mathrm{bb}} = 0.872 \cdot \frac{\int_{\lambda_1}^{\lambda_2} F_\lambda^\uparrow \, \mathrm{d}\lambda}{\int_{\lambda_1}^{\lambda_2} F_\lambda^\downarrow \, \mathrm{d}\lambda} + 0.034 \qquad \text{(cloudy).} \tag{2}$$

A nadir-pointing infrared sensor KT19.85 (Ehrlich et al., 2019) with a field-of-view (FOV) of 2° was installed on both aircraft. The sensor measures the brightness temperature of the surface along the flight track. The instrumental sensitivity covers parts of the atmospheric window between 9.6 μm and 11.5 μm wavelengths. The surface skin temperature can be related to the

brightness temperature of the KT19.85 with uncertainties below ±0.2 K (Stapf et al., 2019).

### 2.2.2 Camera observations

The classification of surface types is based on images taken by different camera systems. Images were partitioned by manually selected thresholds of the three spectral channels in red, green, and blue (RGB) (Perovich et al., 2002; Jäkel et al., 2019). Depending on the illumination conditions, these thresholds were set using color intensity histograms of training samples.

Digital cameras (Canon EOS 1D Mark III and Nikon D5) with fisheye lenses were used by default during most of the flight experiments on both aircraft. Due to the 180° FOV, such cameras cover the entire lower hemisphere that can be directly related to upward irradiance measurements (Jäkel et al., 2019). The angular resolution is less than 0.1°. Geometric, spectral and radiometric calibration were applied to characterize the cameras (Ehrlich et al., 2012; Carlsen et al., 2017; Jäkel et al., 2019). For flight tracks, which were not observed by a fisheye camera due to instrumental failures, images were extracted from video

camera data (miniature camera AXIS P1214-E) with about 1 Hz resolution.

A special version of the scientific Modular Aerial Camera System (MACS) was installed on P6 during MOSAiC-ACA. The sensor head of MACS is equipped with matrix array CCD/CMOS/thermal-infrared cameras looking in nadir direction. The

**Table 2.** Overview of the instrumentation and products of Polar 5 and Polar 6 as used in this study.

| Instrument | ACLOUD | PAMRACMiP | AFLUX | MOSAiC-ACA | HALO-$(\mathcal{AC})^3$ | Measured Quantity and Measurement Frequency | Product |
|---|---|---|---|---|---|---|---|
| CMP-22 pyranometer | P5/6 | P5/6 | P5 | P5/6 | P5/6 | Upward, downward irradiance (20 Hz) | Surface albedo |
| SMART | P5 | P5 | | P5 | P5 | Upward, downward irradiance (20 Hz) | Surface albedo |
| KT19.85 | P5/6 | P5 | P5/6 | P5/6 | P5/6 | Upward brightness temperature (20 Hz) | Skin temperature |
| Fisheye digital cameras | P5/6 | P5 | P5 | P5 | P5/6 | RGB image (4 - 6 sec) | Surface type |
| MACS | | | | P6 | | RGB orthomosaics (4 Hz) | Surface type |

maximum continuous image acquisition rate is four frames per second enabling an overlap of images to produce orthomosaics along the flight path. A summary of the relevant airborne instrumentation is given in Table 2.

### 2.2.3 Satellite observations

Satellite-based surface albedo data were derived from the Land Colour Instrument (OLCI) onboard of Sentinel-3 (Kokhanovsky et al., 2019). OLCI covers the spectral range between 400 nm and 1020 nm distributed over 21 spectral bands. The retrieved surface broadband albedo with a spatial resolution of 6.25 km is a product of the Melt Pond Detector (MPD) algorithm, which has been established for the Medium Resolution Imaging Spectrometer (MERIS) data onboard Envisat (Istomina et al., 2015a) and now has been adapted to OLCI (Istomina et al., 2023). Cloud screening is based on a synergy of OLCI and the Sea and Land Surface Temperature Radiometer (SLSTR). A revised spectral-to-broadband conversion (STBC) approach was developed by Pohl et al. (2020) and is applied here. It calculates the surface broadband albedo $\alpha_{\mathrm{bb}}$ over the wavelength range of 300 – 3000 nm from the spectral surface albedo retrieved at wavelengths $\lambda_j$ = 400, 500, 600, 700, 800, and 900 nm:

$$\alpha_{\mathrm{bb}} = \sum_{j=1}^{6} k_j \cdot \alpha_{\lambda_j} \quad j = 1, 2, 3, 4, 5, 6 \tag{3}$$

The coefficients $k_j$ were derived empirically from spectral and surface broadband albedo measurements over landfast ice, being: 0.9337, -2.0856, 2.9125, -1.6231, 0.675, and 0.0892.

### 2.3 Surface albedo parametrization in HIRHAM-NAOSIM

The surface albedo scheme of the coupled regional climate model HIRHAM–NAOSIM was recently revised based on surface broadband albedo measurements performed during the ACLOUD campaign (Jäkel et al., 2019). The original parametrization of the snow albedo, as described in Dorn et al. (2009), was adapted with respect to illumination dependence and snow property changes in terms of threshold temperatures describing the transition between dry and melting snow.

In general, the surface broadband albedo in an inhomogeneous model grid cell is composed of the sum of the surface subtype

albedo values weighted by the areal fractions ($c$) of the respective surface subtypes of open water (subscript ow) and sea ice (subscript i), where sea ice is further divided into snow-covered ice (subscript s), bare ice (subscript bi), and melt ponds (subscript mp):

$$\alpha = c_{\mathrm{ow}} \cdot \alpha_{\mathrm{ow}} + c_{\mathrm{i}} \cdot \alpha_{\mathrm{i}}$$

$$\alpha_{\mathrm{i}} = c_{\mathrm{s}} \cdot \alpha_{\mathrm{s}} + c_{\mathrm{mp}} \cdot \alpha_{\mathrm{mp}} + c_{\mathrm{bi}} \cdot \alpha_{\mathrm{bi}} \quad .$$

(4)

Note that the surface type "white ice", which is a highly reflective scattering layer on top of melting bare ice (Macfarlane et al., 2023), is not explicitly considered in HIRHAM-NAOSIM. Due to its higher albedo compared to bare ice, white ice is added to the class of snow-covered ice in this work and classified accordingly based on camera observations during the measurement flights. The open water fraction is not parameterized within this surface albedo scheme and is calculated with a separate prognostic equation. The surface albedo of open water is set to a fixed value of 0.1, while the individual subtype albedo of the ice types is assumed to be variable within a given range of the surface skin temperatures ($T_{\mathrm{surf}}$) in units of °C (Køltzow, 2007; Dorn et al., 2009; Jäkel et al., 2019). A temperature range with temperature thresholds $T_{\mathrm{d}}$ was defined within which the surface albedo varies linearly from maximum (dry ice/snow) to minimum values (melting ice/snow). The parameterized surface albedo of the snow-covered ice, for example, is then determined by:

$$\alpha_{\mathrm{s}} = \alpha_{\mathrm{min}} + (\alpha_{\mathrm{max}} - \alpha_{\mathrm{min}}) \cdot f(T_{\mathrm{surf}}) \quad ,$$

(5)

with $f(T_{\mathrm{surf}})$ representing the surface skin temperature-dependent function:

$$f(T_{\mathrm{surf}}) = \min(1, \max(0, T_{\mathrm{surf}}/T_{\mathrm{d}})) \quad .$$

(6)

The same holds for the two other surface types bare ice and melt ponds. Table 3 summarizes the surface albedo ranges of the individual subtypes and threshold temperatures. The areal fractions of the sea ice subtypes used in Eq. (4) is estimated by the

**Table 3.** Minimum and maximum values of the surface albedo for each ice type (snow-covered ice, bare ice, melt ponds) used in the albedo scheme of HIRHAM-NAOSIM.

| Ice subtype | Minimum albedo $\alpha_{\mathrm{min}}$ | Maximum albedo $\alpha_{\mathrm{max}}$ | Threshold temperature $T_{\mathrm{d}}$ (°C) |
|---|---|---|---|
| snow-covered ice (cloudy) | 0.80 | 0.88 | -3.0 |
| snow-covered ice (cloudless) | 0.66 | 0.79 | -2.5 |
| bare ice | 0.51 | 0.57 | -0.01 |
| melt ponds | 0.16 | 0.36 | -2.0 |

snow thickness ($h_{\mathrm{s}}$). For snow-covered sea ice, the fraction is calculated with:

$$c_{\mathrm{s}} = c_{\mathrm{s,max}} \cdot \tanh\left(\frac{h_{\mathrm{s}}}{h_{0.75}}\right)$$

(7)

where $c_{s,max}$ is the maximum snow cover fraction of 1.00 and $h_{0.75} = 0.03$ m giving the snow thickness at which approximately 75 % of the sea ice is covered by snow (Dorn et al., 2009). The melt pond fraction is subject to the restriction that it is not allowed to exceed the fraction of the sea-ice surface not covered with snow (1-$c_s$). It is parameterized by:

$$c_{mp} = \min(1 - c_s, c_{mp,max} \cdot (1 - f(T_{surf}))) \tag{8}$$

with $c_{mp,max}$ being the maximum melt pond fraction of 0.22 as derived from observational data during SHEBA (Køltzow, 2007; Perovich et al., 2002) which agrees with observations made during MOSAiC (Webster et al., 2022). Locally, however, higher melt pond percentages may occur, e.g., on level first-year ice (Istomina et al., 2015b). Finally, the bare ice fraction is calculated as the residual ($c_{bi} = 1 - c_s - c_{mp}$). Note that if the actual ice thickness is lower than 0.25 m, then a linear transition between water and bare ice albedo is applied to account for the transparent behaviour of thin ice (Dorn et al., 2009).

The model output was given with spatial resolution of about 27 km distributed over 200 x 218 grid points on a circum-Arctic domain. For the prognostic variables of the atmospherical model component HIRHAM, a 1 % nudging to reanalysis data of the ERA5 data set (Hersbach et al., 2020) was applied. The HIRHAM-NAOSIM model was run for 2018 covering the time frame of the PAMARCMiP campaign (temporal resolution of 1 hour), and for the entire MOSAiC period (temporal resolution of 3 hours) that includes the time frame of the ground-based measurements from spring to autumn 2020 and the period of the aircraft observations during MOSAiC-ACA.

## 2.4 Methodology for comparison

The outline of the comparison of measured and modeled surface albedo is illustrated in Fig. 2. In the first step the albedo scheme was run offline (i.e. without HIRHAM-NAOSIM) with decoupling the two parametrizations of the subtype surface albedo and subtype surface fraction. Because the subtype surface fraction is parameterized as a function of the snow depth, which was not a measured parameter, we used only the parametrization of the subtype surface albedo along with measured values of the prognostic variable $T_{surf}$ and the measured surface type fractions. The offline evaluation was applied to perform a seasonal comparison between observed and parameterized surface albedo considering data of all aircraft campaigns. In the second step, for an online evaluation, the HIRHAM-NAOSIM output was compared to airborne and ground-based observations. The satellite-based surface albedo derived from Sentinel-3 OLCI was used to characterize the spatial variation on an intermediate grid size scale between local ground-based or aircraft observations and the model output. To match the satellite and model data, all data points of the satellite product, that fall into one single grid point of the model, were area-averaged accordingly. However, since the satellite product can only be derived for cloudless conditions, the comparison is limited to a few cases.

## 3 Results

### 3.1 Spatiotemporal variability of surface types and surface albedo

An overview of the proportions of classified subtypes along the flight tracks of the five campaigns is shown in Fig. 3 as a stacked area plot. The temporal evolution of the surface broadband albedo and surface skin temperature are given in the corresponding

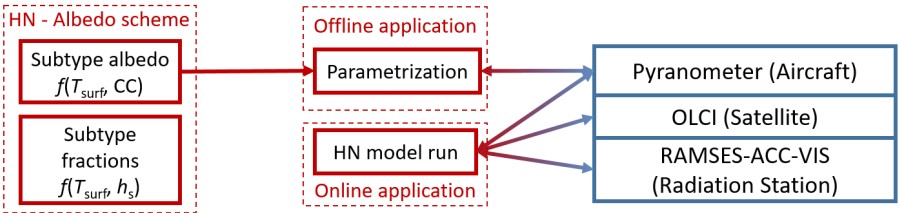

**Figure 2.** Schematics showing the approach for the model-to-measurement comparison. HN stands for HIRHAM-NAOSIM, CC indicates the dependence of the subtype albedo on the cloud coverage.

lower panels. Flight sections in spring (Figs. 3a-f) were mostly carried out over snow or white ice (ice with a highly scattering layer on top) with surface skin temperatures below -15 °C. As expected, the variability of the surface albedo depends on the variability of the surface types within the FOV of the downward-looking pyranometer. However, also flight sections over snow or white ice revealed a spatial albedo variability in the range of up to ±0.1 as an effect of surface roughness and variations in snow grain size. In particular for high SZAs, surface roughness tends to reduce the surface albedo compared to a flat surface depending on the feature orientation with respect to the sun (Larue et al., 2020). Note that longer distances were overflown during PAMARCMiP compared to the other two spring campaigns, which explains the strong variation of $T_{surf}$ (Fig. 3b). The occurrence of open water, either caused by sea ice dynamics or flight sections close to the sea ice edge, leads to a significant increase of surface albedo variability and a decrease of the surface albedo itself down to 0.2. A small percentage of melt ponds was found only in areas with a high fraction of open water (flight on 4 April 2019 during AFLUX) when $T_{surf}$ is close to 0 °C. The onset of melt pond development on sea ice usually starts in summer, as observed at the end of the ACLOUD campaign (26 June) with melt pond fractions of up to 8 % (Fig. 3g). In general, the surface albedo decreases over time as a consequence of an increase of surface grain size and melt pond fraction, which are both related to the increase of skin temperature during ACLOUD (Fig. 3h). In September the overflown surface showed the most variable conditions (Figs. 3i,j). Surface sections during the first flight were dominated by open water with surface skin temperatures being in a similar range as for sea ice (most southern flight track in Fig. 1). Camera images showed that most of the melt ponds were already refrozen and, therefore, classified as bare ice.

Typically, an increase of sea ice fraction is correlated with an increase of the surface albedo. This relation is influenced by the spectral and directional distribution of the incident solar radiation. Compared to cloudless conditions, clouds may cause an increase in surface broadband albedo due to a spectral shift in the incident radiation. The shift is caused by absorption of solar radiation in the near-infrared spectral range. Figure 3k shows the relationship of the sea ice fraction (white ice plus bare ice) and the surface albedo for all campaigns separated into cloudless and cloudy cases. The surface albedo was averaged within each bin of sea ice fraction (bin size of 10 %). In general, we observed a higher albedo for the same amount of sea ice under cloudy conditions than under cloudless conditions. This effect was more pronounced when a high proportion of sea ice was present. For sea ice fractions around 100 % mean surface albedo values of 0.76±0.08 (cloudless cases) and 0.81±0.08 (cloudy cases) were calculated. The variability of the surface albedo at 100 % sea ice cover was further examined with respect to a

potential dependence on the SZA. For cloudless conditions the correlation coefficient of $R = 0.37$ indicates a low correlation between both quantities. In a cloudy atmosphere the incoming radiation is dominated by the diffuse component, which is independent on the SZA as confirmed by a $R$-value of -0.02. Since the majority of measurements was carried out for surface skin temperatures below -15 °C, melting processes can be ruled out as the cause of the surface albedo variability. Only during the summer campaign (ACLOUD), a temperature effect on the magnitude of snow albedo was observed (Jäkel et al., 2019).

## 3.2   Application of surface albedo scheme (offline evaluation)

The surface albedo scheme of the coupled HIRHAM-NAOSIM model was applied to the measurement data of the different aircraft campaigns to evaluate the performance of the surface albedo parametrization for spring, summer, and autumn conditions. Taking the measured subtype fractions into account, the surface albedo $\alpha$ was parameterized following Eq. (4) using surface type specific albedo values defined by Eq. (5). The results separated into cloudy and cloudless cases for spring, summer, and autumn are presented in Fig. 4. The plots show the distributions of the measured and parameterized surface albedo, together with the median value. The parametrization was initially optimized based on the ACLOUD summer campaign data set leading to a reasonable agreement between measurements and parametrization with a root-mean-squared error (RMSE) of 0.05 (Jäkel et al., 2019).

For cloudless conditions, the median values of the measured surface albedo are well represented by the parametrization. A different picture is revealed for cases where clouds are present. While in summer the modeled median surface albedo deviates from from the measured value by only 0.01, we observe an overestimation of the modeled median surface albedo of 0.06 in spring. The increase of the surface albedo caused by clouds in summer is much less pronounced in the spring measurements. As this cloud effect depends on the cloud absorption of the downward irradiance in the NIR spectral range, it is assumed that the generally optically thinner clouds in spring do not alter the spectral surface albedo in the same magnitude as the optically thicker clouds in summer. Therefore, we argue that the cloud parameter in the surface albedo scheme, that was defined for summer conditions, leads to the overestimation of modeled surface albedo in spring. The distributions shown for autumn (MOSAiC-ACA) are primarily affected by the surface sampled. While measurements under cloudless conditions mostly coincided with flight sections over areas with a high fraction of open water, most of the cloudy cases were sampled over compact ice (see Fig. 3i), which justifies the large difference between the two median values of the cloudless and cloudy distributions. The modes representing the measurements over sea ice indicate a higher parameterized surface albedo than derived from the measurements. That is possibly caused by the presence of refrozen melt ponds, which were classified as bare ice. Compared to bare ice of considerable thickness that occurs after snow melt, refrozen melt ponds have little ice thickness on top, resulting in a dark appearance and a lower surface albedo. The model parameters of bare ice may not properly represent such thin ice layers, leading to an overestimation of parameterized surface albedo. In general, however, the distributions were reproduced by the parametrizations for all seasons using the measured sea ice fractions.

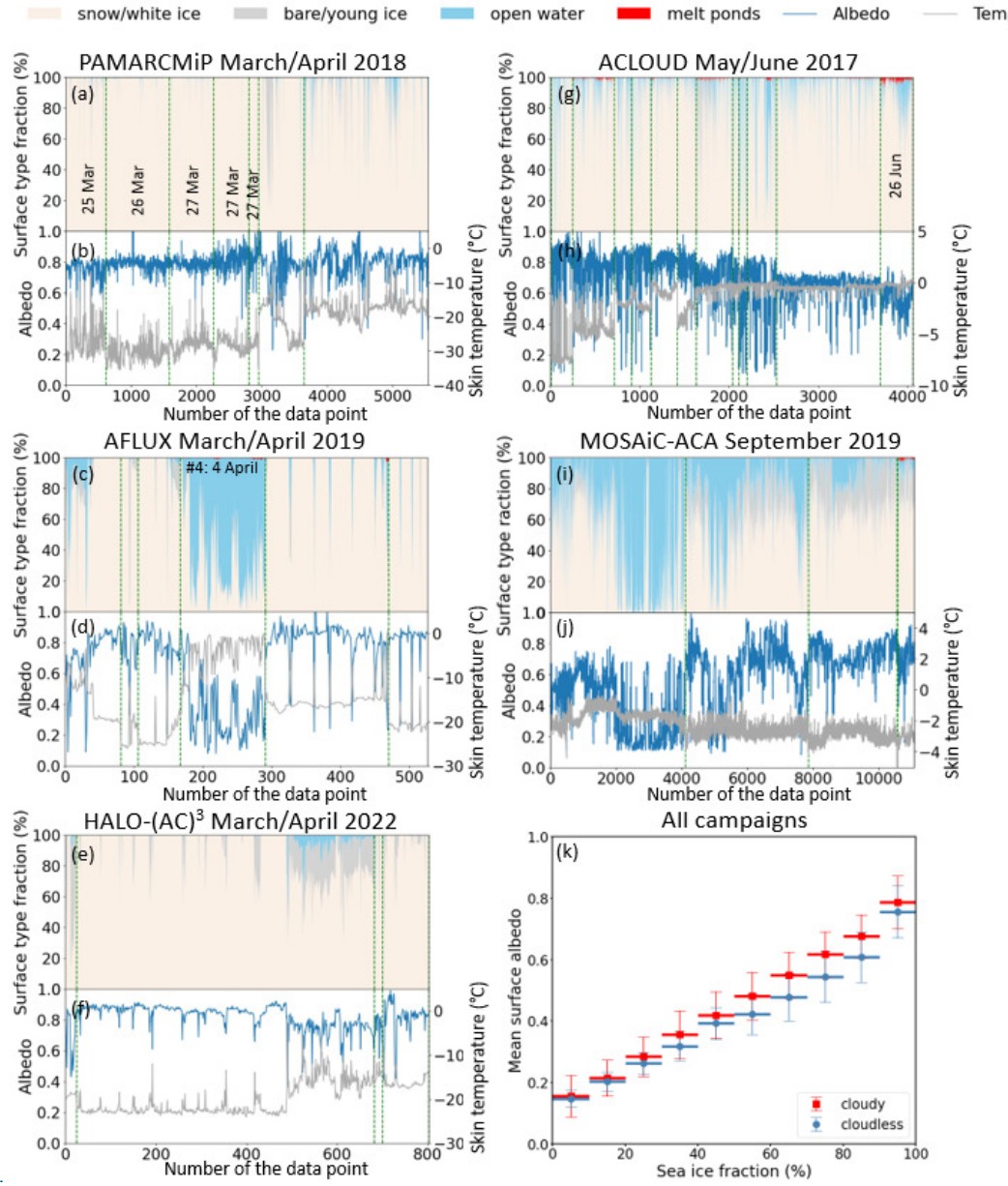

**Figure 3.** (a) - (j) Temporal changes in surface types, surface albedo (blue lines; left y-axis) and surface skin temperature (grey lines; right y-axis) for all five flight campaigns. The proportions of surface types are presented as a stacked area plot to identify the predominant subtypes. Vertical green lines separate the individual flight days. Dates given in the panels are explicitly mentioned in the text. (k) Averaged surface albedo as a function of sea ice fraction (bin size of 10 %), separately for cloudless and cloudy conditions. The standard deviation of the averages is represented by thin vertical bars.

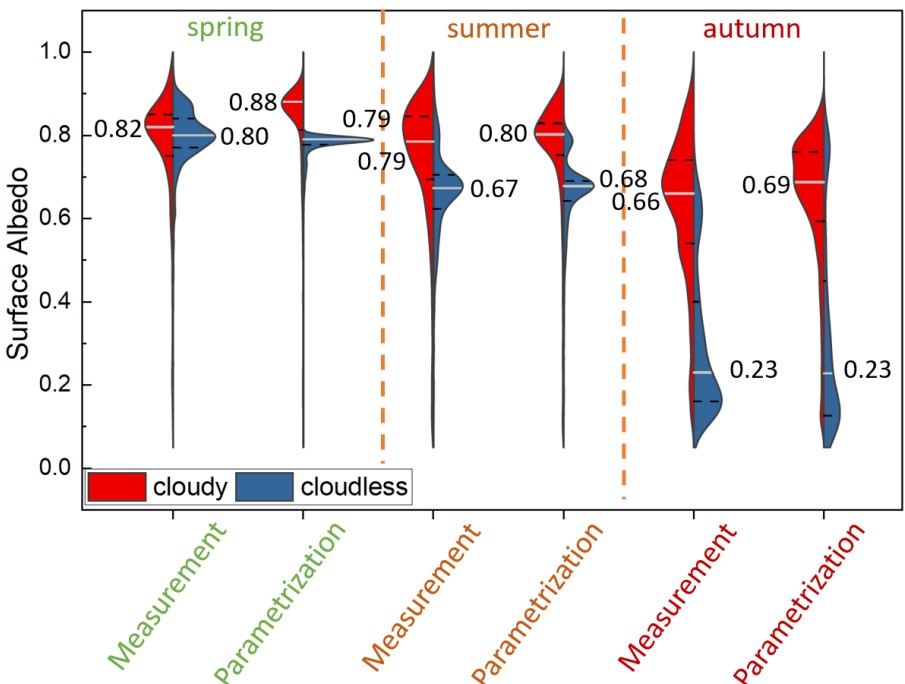

**Figure 4.** Distributions of measured and modeled surface albedo separated in cloudy (red distribution) and cloudless (blue distribution) cases for the seasons spring, summer, and autumn. The median value (also indicated by the white line) is given together with the first and third quartiles (black dashed lines).

## 3.3 Application of the HIRHAM-NAOSIM model (online evaluation)

### 3.3.1 Spring case - PAMARCMiP

For spring conditions, HIRHAM-NAOSIM was applied for the time frame of the PAMARCMiP campaign. As an example, the spatial distribution of the modeled surface albedo for the PAMARCMiP area is shown in Fig. 5a. The position of the ice edge can be clearly identified by the sharp gradient of the surface albedo in the lower right corner of this panel. Aircraft-based photos of the surface showed a few refrozen leads along the flight path, but most of the flight sections were carried out over dense drift ice, far away from the MIZ. Despite most of the ocean being completely covered by sea ice, greater variability is observed than in the model results, as shown by the color-coded dots depicting aircraft measurements for each flight day. Small scale variations arising from surface structure of deformed sea ice, as observed by an airborne laser scanner (Jäkel et al., 2021), cannot be resolved by the model. In contrast, the satellite-based product accounts for variations due to surface roughness (Fig. 5b). A more homogeneous surface albedo was derived for the area north of 82° N latitude. Since surface observations by OLCI are restricted to cloudless scenes, the MPD satellite surface albedo product does not cover the entire area. Therefore, the comparison between satellite-based and modeled surface albedo is limited to the area of data points that is shown in Fig. 5b.

Fig. 5c illustrates the distributions of the different surface albedo products. On the left side, the areal comparison of the MPD and HIRHAM-NAOSIM product is shown. As shown in Fig. 5b, we observe greater variability in the higher resolution satellite data. However, the median values are similar (0.85, 0.86). The smaller second modeled mode (0.77) can be attributed to grid points with a low modeled cloud coverage, hence the snow-covered ice parameter representing cloudless conditions was applied. As the modeled surface albedo depends on cloud cover, the representation of the clouds in the model must be taken into account to evaluate the modeled surface albedo. While the aircraft and satellite observations showed mostly cloudless conditions, the model calculated a cloud cover of about 100 % in most areas. Based on these results, it can be assumed that the match between satellite- and model-derived surface albedo medians results from the compensation of two opposite model biases: the overestimation of modeled cloud coverage, which caused a positive bias in modeled surface albedo, was compensated by a negative bias in modeled cloudless surface albedo. The two distributions of the airborne and the spaceborne surface albedo are shown on the right side of the panel in Fig. 5c. For that, the aircraft measurements were area-averaged with respect to the grid size of the satellite product. The deviation of 0.1 in the median value indicates an overestimation of the satellite product. We applied the spectral-to-broadband conversion of the MPD algorithm (Eq. 3) to the spectral surface albedo measurements of the SMART instrument to rule out larger uncertainties in this conversion, as they might occur in these low-sun conditions during PAMARCMiP. Using the spectral surface albedo of SMART at the six wavelengths together with the spectral weighting coefficients we calculated only a small bias to the broadband measurements with a RMSE of 0.02 and similar mean values of 0.74. Therefore, the spectral-to-broadband conversion can be excluded as a reason for the positive bias of the satellite-based surface albedo.

The temporal variation of the modeled surface albedo is illustrated in Fig. 5d. Each individual line represents the time series of the area-averaged surface albedo for one of the seven overflown areas. In addition, the mean measured surface albedo (including standard deviation) on the corresponding day is shown. Apart from the most southern region overflown on 3 April 2018, no significant change of the surface albedo within the time frame of the campaign were simulated. Short-term variations can be attributed to changes in modeled cloud cover, while larger temporal variations are correlated to the modeled sea ice coverage. The albedo time series of the area overflown on April 3, 2018 shows a pronounced albedo minimum in late March. This is related to a modeled minimum sea ice cover of 86 % for the period shown, which is probably due to ice dynamics. In general, the measured surface albedo shows much greater spatial variability but smaller averaged values than the model. This positive bias of modeled surface albedo cannot be explained by a lower sea ice coverage modeled with HIRHAM-NAOSIM. In fact, the observed sea ice cover averages 99 %, while modeled sea ice cover ranges from 94 % to 99 %. Rather, the biases in the modeled cloud fraction may explain some of the discrepancy between modeled and measured surface albedo. In particular, the first three flights were conducted under cloudless conditions, which would lower the surface albedo from more than 0.85 to 0.76 assuming 95 % sea ice coverage and a snow albedo of 0.79 (Table 3). At least on the following measurement days, the modeled surface albedo is within the standard deviation of the measured surface albedo. However, the area-averaged surface albedo deviates by up to 0.1.

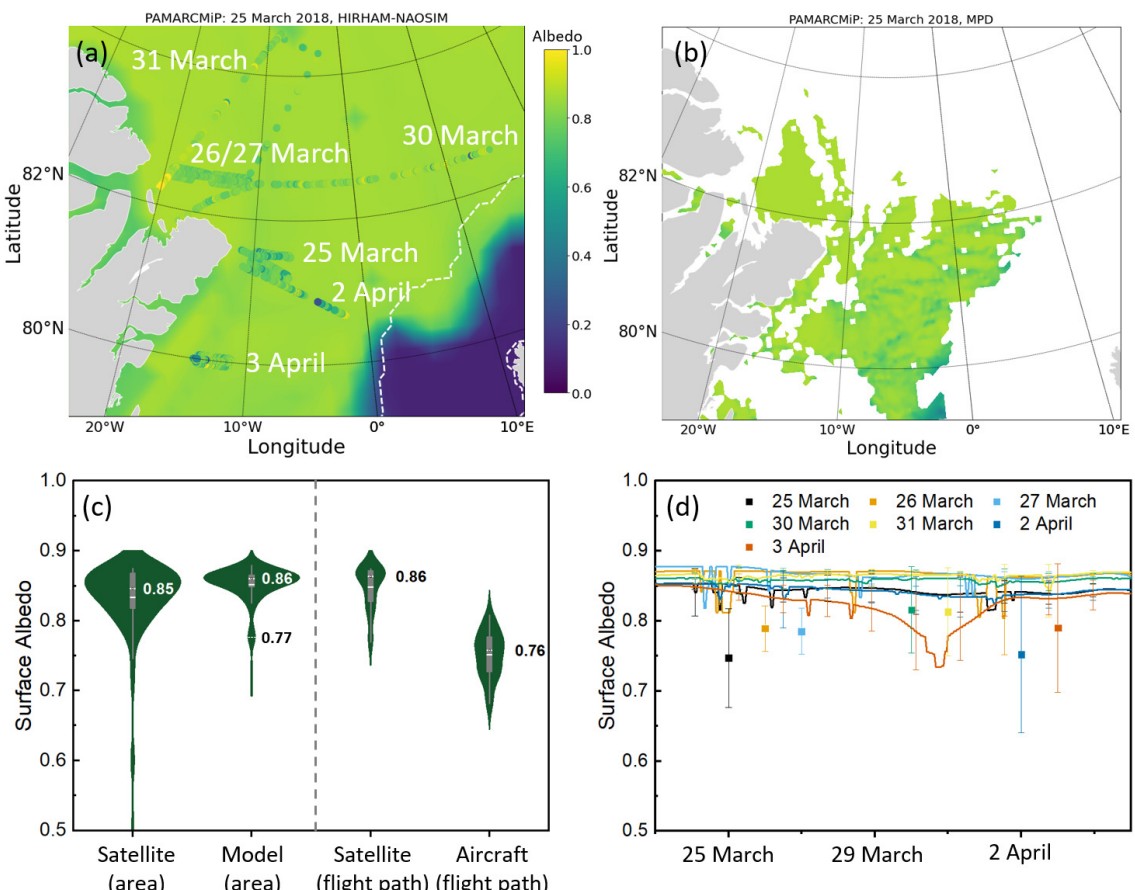

**Figure 5.** (a) Map of the surface albedo as modeled by HIRHAM-NAOSIM for 25 March 2018 (PAMARCMiP). Aircraft observations are plotted as color-coded dots indicating the measured surface albedo for the individual flight days. The white dashed line indicates a SIC of 15 % derived from AMSR observations. (b) Surface albedo under cloudless conditions derived from the OLCI measurements by the MPD retrieval for 25 March 2018 (same color code as in (a)). (c) Distributions of satellite, model, and aircraft surface albedo data. Left: statistics of model and MPD retrieval for the area that is covered by the satellite data shown in b), right: comparison along the flight path only. (d) Time series of modeled mean surface albedo for the PAMARCMiP period for the areas covered by the individual flights. The aircraft measured mean surface albedo (single squares) and standard deviation (vertical bars) are given together.

### 3.3.2 Autumn case - MOSAiC-ACA

HIRHAM-NAOSIM was further applied for the period of the MOSAiC campaign 2020. The accompanying aircraft observa-
tions in autumn during MOSAiC-ACA revealed four days with measurements of the surface albedo as depicted in Fig. 6a. Other than during PAMARCMiP, the flights were performed over a strongly heterogeneous surface in the MIZ. Again, the modeled sea ice edge can be estimated from the spatial distribution of surface albedo, as the transition to the blue colored areas coincides with the sea ice edge zone. However, AMSR observations of the SIC show a more eastward shift of the sea ice edge

compared to the model as illustrated by the 15 % iso-line of SIC in Fig. 6a. This could partly explain the difference between the
modeled and measured surface albedo. The modeled albedo map shows a negative bias compared to the measurements along
the flight path (overlaying brighter points in Fig. 6a), especially for the flights on 8 and 13 September. The corresponding time
series of the area-averaged modeled surface albedo for the four flight regions are shown in Fig. 6b. Compared to the spring data
set, a greater spatiotemporal variation is observed which is mainly driven by the variation of surface type distribution. Since
HIRHAM-NAOSIM mostly simulated a cloud coverage of 100 %, the variability of the surface albedo cannot be attributed to
the use of different parametrizations for cloudy and cloudless conditions.

The measured area-averaged surface albedo shows best agreement for the region overflown on September 2, although the sur-
face albedo along the northernmost section of this flight path was partly overestimated by the model. During the flight carried
out on 7 September 2020, the area-averaged surface albedo is slightly underestimated by the model, but is still within the
range of standard deviation of the measurements. This differs from the results of the two last flights, which show a significant
negative bias of the modeled surface albedo. Both measurements and model revealed a similar cloud coverage. This suggests
that especially the parametrization of the surface types affects the representation of the modeled surface albedo.

Since the model assumes a sea ice edge closer to the areas observed by the aircraft, the modeled fraction of open water is signif-
icantly higher than the measurements show. To exclude the open water fractions, we only considered the three sea ice subtypes
and scaled them such they sum up to a fraction of one. This makes them more comparable to the modeled sea ice fractions $c_{\mathrm{mp}}$,
$c_{\mathrm{s}}$, and $c_{\mathrm{bi}}$ (Eq. (6)). We further reduced the data sets, where $c_{\mathrm{ow}}$ exceeds 0.8. Table 4 summarizes the area-averaged surface
type fractions as derived from the model and the aircraft observations. Melt ponds do not affect either the modeled or measured
fractions. The relationship between snow depth and $c_{\mathrm{s}}$ (Eq. (7)) leads to an underrepresentation of snow-covered ice for all
days, because of either insufficient modeled snow depth or the relationship itself. Unfortunately, snow depth observations are
not available to look into the cause of the differences. However, the spread of the modeled fractions is significantly larger than
those derived from camera observations due to the more heterogeneous surface conditions at the modeled sea ice edge.

**Table 4.** Area-averaged surface type fractions of snow-covered ice ($c_{\mathrm{s}}$), bare ice ($c_{\mathrm{bi}}$), melt ponds ($c_{\mathrm{mp}}$), and open water ($c_{\mathrm{ow}}$) derived from
the camera classification and modeled by HIRHAM-NAOSIM. The standard deviation demonstrates the spatial variation.

| Day | $c_{\mathrm{s}}$(measured) | $c_{\mathrm{s}}$(modeled) | $c_{\mathrm{bi}}$ (measured) | $c_{\mathrm{bi}}$ (modeled) | $c_{\mathrm{mp}}$ (measured) | $c_{\mathrm{mp}}$ (modeled) |
|---|---|---|---|---|---|---|
| 2 September | 0.85±0.06 | 0.61±0.40 | 0.14±0.06 | 0.37±0.37 | 0.00±0.00 | 0.03±0.04 |
| 7 September | 0.75±0.02 | 0.19±0.14 | 0.25±0.03 | 0.81±0.14 | 0.00±0.00 | 0.00±0.00 |
| 8 September | 0.70±0.04 | 0.51±0.21 | 0.30±0.04 | 0.45±0.15 | 0.00±0.00 | 0.04±0.06 |
| 13 September | 0.77±0.01 | 0.70±0.12 | 0.21±0.01 | 0.30±0.12 | 0.01±0.00 | 0.00±0.00 |


### 3.3.3 Polar day time series - MOSAiC

During MOSAiC, the seasonal evolution of the surface albedo was measured by autonomous radiometers. In this study, data
from one of the RAMSES stations (2020R12, following the notation of Tao et al., 2023) were used. 2020R12 was deployed

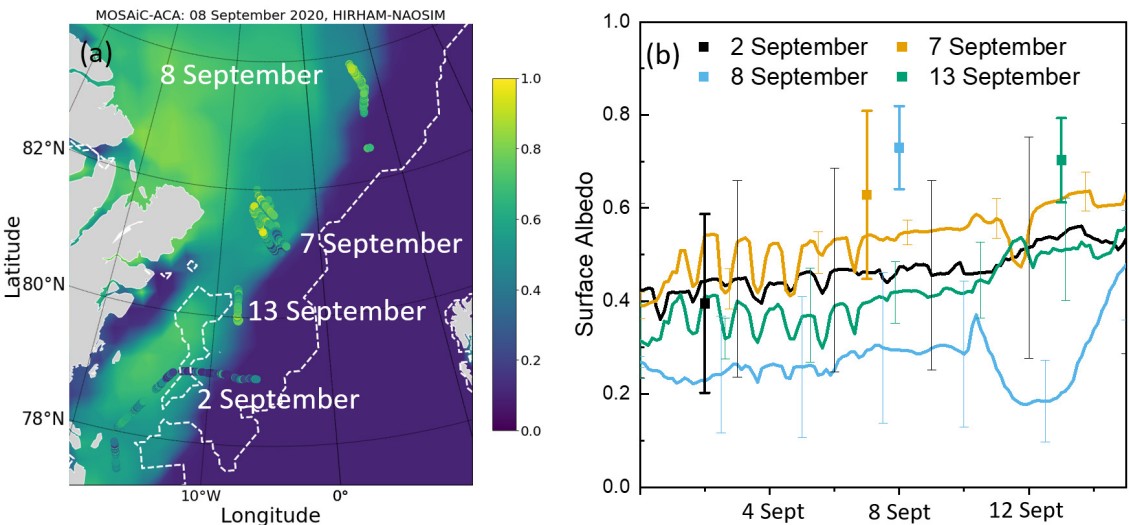

**Figure 6.** (a) Map of the surface albedo as modeled by HIRHAM-NAOSIM for 8 September 2020 (MOSAiC-ACA). Aircraft observations are plotted as color-coded dots indicating the measured surface albedo for the individual flight days. The white dashed line indicates a SIC of 15 % derived from AMSR observations. (b) Time series of modeled mean daily surface albedo for the MOSAiC-ACA period for the areas covered by the individual flights. The standard deviation of the area average is represented by thin vertical bars. The measured mean surface albedo and standard deviation are shown similar as in Fig. 5d.

on second year ice at site L3 of the MOSAiC Distributed Network (Nicolaus et al., 2022). This data set provides almost
continuous time series of irradiance measurements between April 24 and August 7, 2020, which allow to observe the transition
from dry to wet snow during the onset of melting. We applied the two corrections according to Eqs. (1) and (2) to the ground-
based observation of the autonomous radiometers. The time series of original and corrected measured surface albedo (April to
August) are shown in Fig. 7a. As the radiometer data were not filtered with respect to the atmospheric conditions, we assume
that the two time series, representing either cloudless or cloudy conditions, indicate the range of the surface broadband albedo.
The plot shows several characteristics of the melting season as discussed in Tao et al. (2023). Prior to May 26, the surface was
covered with dry snow resulting in the highest surface albedo. With the onset of melting, an initial melt pond formed directly
under the radiation sensor, so that a first minimum of surface albedo was observed on May 29. The snowfall caused the surface
albedo to increase thereafter, but not to the earlier level, as wet snow condition prevailed instead. A second major melt pond
event was observed with a minimum surface albedo on June 25. The later increase of surface albedo is related to melt pond
drainage. After that, the surface was dominated by the surface scattering layer (SSL).

Satellite-based surface albedo data were available for five cloudless days during the period of ground-based observations. The
data were averaged over the area corresponding to the extent of the HIRHAM-NAOSIM grid pixel covering the radiometer
site. For the two days June 5 and 6, when the melt pond was covered by new snow, the satellite-based surface albedo exceeds
the ground-based values (Fig. 7a). On June 21 and 22, satellite and ground-based measurements showed a similar mean surface
albedo of 0.69. For the observed cloudless conditions, Eq. (1) can be applied to correct the radiometer measurements. Largest

differences were found after the drainage of the observed melt pond on June 30. Here, the radiometer measurement exceeds the satellite-based surface albedo product by more than 0.1. One can assume that the local observation was not representative for the larger area observed by the satellite, which captured a higher fraction of melt ponds with a lower surface albedo.

The time series of the modeled surface albedo by HIRHAM-NAOSIM has three major phases. The first ends with the onset of melting similar to the ground-based measurements, on May 26.While the radiometer measurements showed a decrease in surface albedo due to a first melt pond event, the modeled albedo only decreased due to the transition to wet snow. Melt ponds were not modeled at this stage as can be seen in Fig. 7b. In fact, snow covered ice was the dominant surface type fraction. Three days after the observed formation of the second melt pond, a significant change of surface type fractions was also modeled for June 28, 2020. Within two days, pond formation started simultaneously with the transformation of snow covered ice to bare ice due to snow melt. The timing of the second melt pond formation was well simulated by the model. After the formation of melt ponds, however, the modeled surface albedo was significantly underestimated compared to the observations by the satellite and the ground-based RAMSES station. The modeled surface albedo remains on the low level ($\alpha \approx 0.4$) after June 28, while the measured surface albedo ($\alpha > 0.6$) increases again due to surface drainage. The MPD OLCI satellite retrieval also determines the melt pond fraction, which was about 25 % on June 30 (Niehaus et al., 2023) and thus higher than modeled melt pond fraction (20 %). We also assume that the predominantly modeled bare ice fraction with its low surface albedo contributes to the albedo model bias. In the field, however, the surface albedo of the melting ice remained relatively high due to the presence of a brighter SSL, which is not taken into account in HIRHAM-NAOSIM.

## 4 Effect of surface albedo bias on net irradiance

### 4.1 HIRHAM-NAOSIM model results

The net solar irradiance at the surface is defined as the difference of downward and upward irradiance:

$$F_{\text{net}} = F^{\downarrow} - F^{\uparrow} \quad . \tag{9}$$

The difference of the modeled and measured net irradiances is calculated to estimate the impact of the model bias for the solar radiative energy budget:

$$\Delta F_{\text{net}} = F_{\text{net,model}} - F_{\text{net,meas}} \quad . \tag{10}$$

Based on the model results introduced in Sect. 3.3, we compared the measured and modeled net irradiance for all model grid points that covered the selected flight tracks during the PAMARCMiP and MOSAiC-ACA campaign. Figure 8 shows the scatterplot of both net irradiances. The corresponding standard deviation illustrates the variability of $F_{\text{net}}$ with a maximum of about $60\,\text{W}\,\text{m}^{-2}$. The correlation $R$ between the measured and modeled net irradiances is 0.80 and the RMSE of the model is $30.2\,\text{W}\,\text{m}^{-2}$, with deviations increasing accordingly for larger differences between measured and modeled surface albedo.

The $F_{\text{net}}$-differences between measurement and model depend not only on $\Delta\alpha$, but we must also take into account the difference in the downward irradiance ($\Delta F^{\downarrow}$). A negative $\Delta F^{\downarrow}$ (smaller symbols in Fig. 8) may occur when the modeled extinction

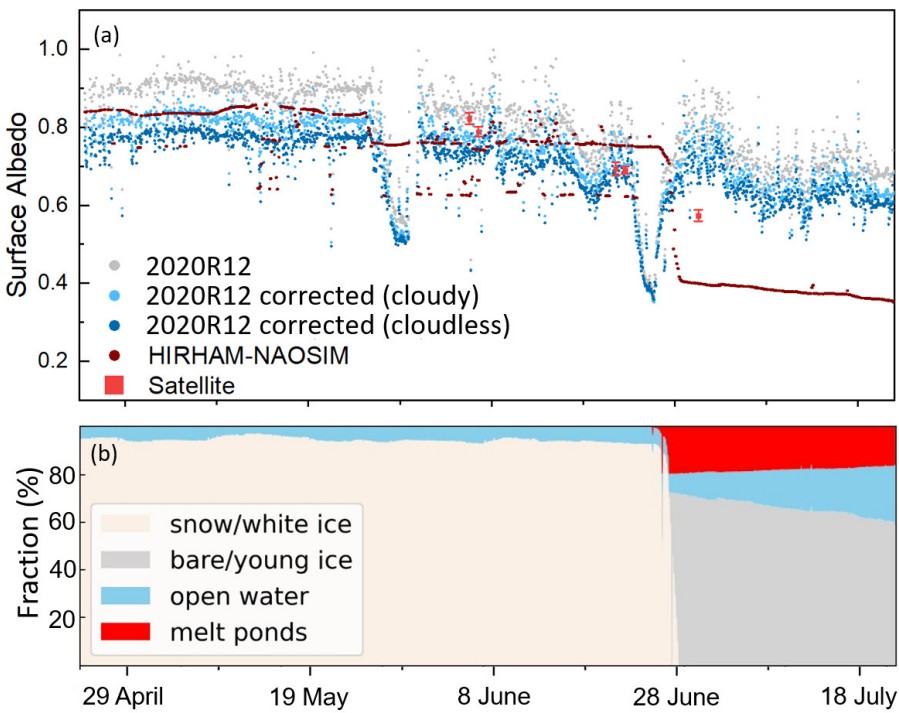

**Figure 7.** (a) Time series of surface broadband albedo (original and corrected) derived from radiometer measurements and HIRHAM-NAOSIM modeling during MOSAiC 2020. Short-term variations in modeled surface albedo are attributed mainly to changes in cloud cover. Area-averaged satellite-based OLCI MPD retrieval results covering the area of the model grid cell are shown for five cloudless days (red symbols). (b) Temporal evolution of surface type fractions calculated by HIRHAM-NAOSIM.

of $F^\downarrow$ caused by modeled clouds is higher than an observation would show. This is especially the case when cloudless situations were observed but not modeled. It would lead to an underestimation of the modeled net irradiance, assuming the same surface albedo. In fact, a mean negative bias of the modeled $F^\downarrow$ (mean $\Delta F^\downarrow = -31 \, \mathrm{W \, m^{-2}}$) was found, which can be related to
an overestimation of the modeled cloud cover. However, the downward irradiance itself also depends on the surface albedo. In particular, $F^\downarrow$ under cloudy conditions is enhanced for brighter surfaces due to multiple-scattering between surface and cloud base. A positive surface albedo bias would lead to a positive bias in $F^\downarrow$, assuming a similar cloud representation. On average $\Delta \alpha$ was 0, indicating a small effect of surface albedo on the modeled $F^\downarrow$.

Overall, both cloud properties and surface albedo must be well represented for modeling net irradiance correctly. To estimate
whether the representation of clouds or the surface albedo potentially contribute more to the uncertainty of $F_{\mathrm{net}}$, we calculated the standardized regression coefficients. Such standardization is useful as the parameters are expressed in different units. HIRHAM-NAOSIM provides the total cloud water path (CWP) as a measure of the cloud microphysics. To account for the available incident radiation we consider also the SZA as a third parameter. The standardized regression coefficients $\beta_j$ with $j$ being either $\alpha$, CWP, or SZA are calculated directly from the unstandardized regression coefficient $b_j$ between $F_{\mathrm{net}}$ and the

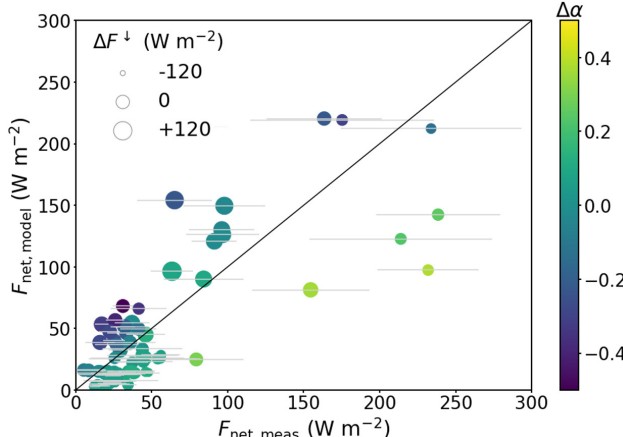

**Figure 8.** Scatterplot of net irradiance based on measured and modeled surface albedo covering the flights performed during PAMARCMiP and MOSAiC-ACA. The horizontal bars indicate the standard deviation of the averaged measured $F_{\text{net}}$. Color code gives the surface albedo difference ($\Delta\alpha = \alpha_{\text{model}} - \alpha_{\text{meas}}$) and the symbol size the difference of the modeled and measured downward irradiance.

variables and the standard deviations ($\sigma$) of the variables:

$$\beta_j = b_j \cdot \frac{\sigma_{\text{j}}}{\sigma_{\text{F}_{\text{net}}}} \quad . \tag{11}$$

Meaning that a change of one standard deviation in one of the parameters is associated with a change of $\beta$ standard deviations of $F_{\text{net}}$, so that the more important variable will have the maximum absolute value of $\beta_j$. For the analyzed cases during MOSAiC-ACA and PAMARCMiP we found the strongest impact of the surface albedo ($\beta_\alpha$ = -0.80), and less impact of the

CWP ($\beta_{\text{CWP}}$ = -0.38) and SZA ($\beta_{\text{SZA}}$ = -0.23). This highlights the importance of a reliable surface albedo parametrization for modeling a realistic surface energy balance. However, we expect a seasonal dependence of the standardized regression coefficients. According to Eq. (11), a stronger variability of the individual parameters contributes to a higher magnitude of $\beta_j$. In summer, for example, clouds tend to have a higher cloud water path with greater variability, while the surface albedo reaches its minimum. Therefore, it is assumed that the contribution of the surface albedo bias to the $F_{\text{net}}$ uncertainty is reduced, while

the model representation of cloud properties becomes more important compared to the two periods shown in this study .

### 4.2   Offline evaluation results

In contrast to the study of the HIRHAM-NAOSIM results, the application of the offline evaluation allows to consider dependencies of the $F_{\text{net}}$ bias for the comparison of the parametrization with the airborne measurements. The measured subtype fractions were used to identify only the influence of the bias of the parameterized surface albedo on $F_{\text{net}}$, without having to

consider the uncertainties of the subtype fraction parametrization. The net irradiance was determined along the flight path for seven selected days during all five flight campaigns, covering cloudy and cloudless conditions. Radiative transfer simulations (Appendix A) were performed for these cases using the measured and parameterized surface albedo. In this way, the sensitivity

of net irradiance to surface albedo was quantified under the same predefined atmospheric condition. These conditions matched the measurements made during the selected flights (see Appendix A). For cloudless conditions broadband upward and downward irradiances were simulated, so that the direct impact of surface albedo can be derived from the difference of the resulting net irradiance, according to Eq. (10). For cloudy conditions the setup of the radiative transfer model required information of the cloud microphysical properties. We estimated these profiles along the flight path as follows: Where appropriate, we used profile in situ measurements of the liquid or ice water content and particle size to define a standard cloud profile. In a second step this standard profile was adjusted by scaling the water content of the profile so that the measured and the simulated downward irradiance were matching at each measurement point along the flight track. This provides an estimate of the cloud optical depth (COD). Radiative transfer simulations were then performed using this scaled cloud profile and the parameterized surface albedo to derive $F_{\text{net,model}}$. A table summarizing the corresponding microphysical profiles can be found in Appendix A.

Figure 9a shows a scatterplot of the net irradiances that were derived from the measured and parameterized surface albedo. A significant smaller spread between measured and parameterized $F_{\text{net}}$ with $R = 0.97$ and RMSE $= 13.5\,\text{W}\,\text{m}^{-2}$ is obtained. We identify two clusters. The first one represents all spring/autumn cases, and data derived for cloudy conditions in summer ($F_{\text{net}} < 100\,\text{W}\,\text{m}^{-2}$). As an effect of a low downward irradiance, and a high surface albedo, the lowest $F_{\text{net}}$-values were derived for spring cases under cloudy conditions that are dominated by a high fraction of dry snow surfaces. The second cluster ($F_{\text{net}} : 150\text{--}350\,\text{W}\,\text{m}^{-2}$) indicates the cloudless cases in summer that are obtained at a lower SZA ($56°$ - $66°$) and low surface albedo due to wet snow. Figure 9b illustrates the dependence of $\Delta F_{\text{net}}$ on $\Delta\alpha$ and SZA. The linear relationship between $\Delta F_{\text{net}}$ and $\Delta\alpha$ for similar atmospheric conditions results directly from the correlation of surface albedo and upward irradiance. A positive bias of the parameterized surface albedo leads to a higher upward irradiance and consequently results in a lower $F_{\text{net}}$ compared to the measurements. The maximum impact of the albedo bias on $\Delta F_{\text{net}}$ is derived for cloudless summer conditions ($\Delta F_{\text{net}} = \pm 80\,\text{W}\,\text{m}^{-2}$). For the same range of $\Delta\alpha$ in spring, $\Delta F_{\text{net}}$ is found to be less than half of its magnitude in summer ($\Delta F_{\text{net}} = \pm 35\,\text{W}\,\text{m}^{-2}$). This means that the bias of modeled surface albedo can have greater effects on the simulated net solar irradiance at the surface in summer compared to spring. In spring, however, we observed from the flight measurements an increased albedo bias with a wider distribution ($\Delta\alpha = 0.02 \pm 0.07$) than in summer ($\Delta\alpha = 0.00 \pm 0.04$). The deviation from the linear relationship between $\Delta F_{\text{net}}$ and $\Delta\alpha$ at a similar SZA can be attributed to different cloud conditions. We used the estimated cloud optical depth to illustrate the cloud impact on $\Delta F_{\text{net}}$. Fig. 9c shows the frequency distribution of $\Delta F_{\text{net}}$ of all analyzed cases separated into three cloud classes. Accordingly, the cloudless cases comprise the largest range of values with a low mean positive bias (median: $2.5\,\text{W}\,\text{m}^{-2}$). The $F_{\text{net}}$ bias for cases with optically thicker clouds (COD $> 5$) is distributed around its median value of $0.1\,\text{W}\,\text{m}^{-2}$, showing the narrowest distribution (interquartile range: $10.4\,\text{W}\,\text{m}^{-2}$). A clear negative $F_{\text{net}}$ bias (median: -6.4 $\text{W}\,\text{m}^{-2}$) is observed for optically thin clouds (COD $< 5$), which results from the systematic overestimation of the cloud enhancement effect for $\alpha_{\text{model}}$ in case of optically thin clouds. A better description of the surface albedo dependence on the cloud property is required to overcome this systematic effect.

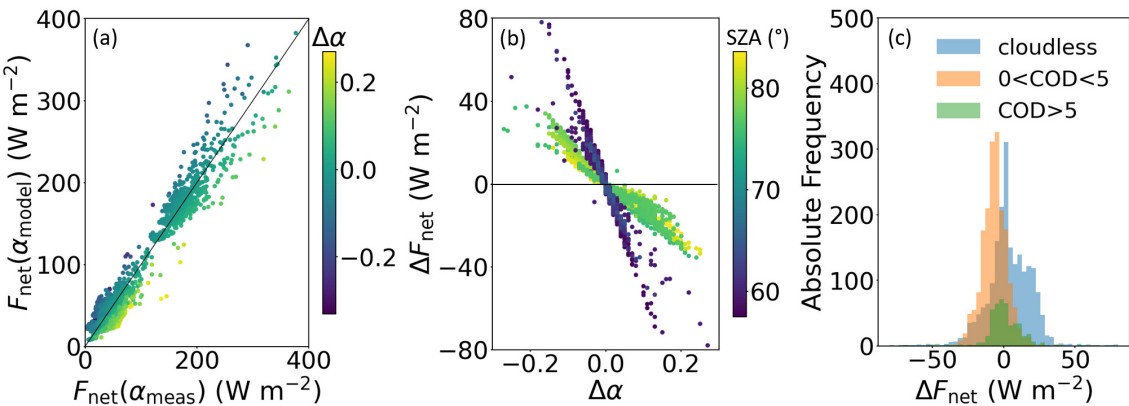

**Figure 9.** (a) Scatterplot of net irradiance based on measured and parameterized surface albedo covering flights performed in spring, summer and autumn. Color code gives the surface albedo difference ($\Delta\alpha = \alpha_{\mathrm{model}} - \alpha_{\mathrm{meas}}$). (b) Difference of net irradiance with parameterized surface albedo and net irradiance with measured surface albedo as function of surface albedo difference. Colors indicate the solar zenith angle. (c) Frequency distribution of $\Delta F_{\mathrm{net}}$ separated into three cloud classes: cloudless, thin clouds with cloud optical depth (COD) lower than 5, and clouds with COD larger than 5.

## 5  Summary and conclusions

In this study, an extensive data set of aircraft measurements of the surface albedo was used for evaluating the parameterized surface albedo from the coupled regional climate model HIRHAM-NAOSIM applied in the Arctic. The measurements were collected during five field campaigns in the European Arctic in different seasons between 2017 and 2022. Different approaches were applied to compare the measured and parameterized surface albedo. In an offline evaluation measured surface type fractions were used to identify deficiencies of the surface albedo parametrization itself, whereas the direct application of the HIRHAM-NAOSIM model (online evaluation) allowed an evaluation of the two components of the surface albedo scheme (subtype albedo and subtype fraction parametrization).

A regression analysis of the relationship between measured sea ice fraction and measured surface albedo confirmed the increase of the surface broadband albedo in the presence of clouds. We found that the dry snow albedo assumed in HIRHAM-NAOSIM for cloudless cases (0.79) was well in agreement with the airborne measurements (0.76±0.08), while for cloudy conditions the assumed albedo for dry snow in the model was slightly overestimated (0.88 vs. 0.81±0.08). However, the measured surface albedo of dry snow is at the lower limit compared to literature data where surface albedo ranges between 0.8 and 0.9 (Perovich et al., 2002; Light et al., 2022).

For the offline evaluation, the parametrization reproduced the measured surface albedo distributions for all seasons, in particular for cloudless conditions. In contrast to the parametrization, however, the measured increase of the surface broadband albedo under cloudy conditions is much less pronounced in spring than in summer, which is attributed to differences in cloud optical thickness. In the absence of a waveband-dependent albedo parametrization, the consideration of a simple cloud dependence in the broadband albedo parametrization is able to mimic the cloud effect on surface albedo reasonably. The cloud effect

might be further improved by a more sophisticated functional dependence on cloud cover or cloud water content, rather than a pure distinction between cloudy and cloudless conditions. Such an approach was proposed by Gardner and Sharp (2010), who developed a snow albedo parametrization as a function of cloud optical depth. The application of this parametrization (Eq. (11) in Gardner and Sharp, 2010) has shown some improvements in the offline evaluation for cases with optically thin clouds. However, a comprehensive online evaluation is difficult, because this approach uses COD, a variable which is usually not available, neither in HIRHAM-NAOSIM nor in most other climate models.

The comparison of the HIRHAM-NAOSIM simulations with the PAMARCMiP data showed that the modeled surface albedo was affected by biases in the modeled cloud cover (cloudy instead of cloudless). For days with correctly modeled cloud cover, the modeled surface albedo was within the standard deviation of the measured values. This demonstrates that reliable cloud cover modeling is needed to properly account for the dependence of surface albedo on clouds. For the autumn MOSAiC-ACA campaign, which was characterized by much larger variation in surface types, the error of modeled surface albedo can primarily be attributed to uncertainties in the surface type parametrization.

The comparison with ground-based observations from one of the drifting radiation stations during MOSAiC showed that the onset of the melt season and the drop in surface albedo due to the transition from dry snow to wet snow were well reproduced. Larger surface albedo differences (more than 0.1) were obtained after the drainage of the observed melt ponds at the end of June. From this time on, the largest discrepancies between observations, including satellite-based surface albedo measurements, and model results were found. This phase of the melt season was not well reproduced by the model. In particular, the surface albedo after disappearance of the snow cover is underestimated. This is due to the fact that the model assumes bare ice instead of a surface scattering layer (SSL), which emerges at the top of the melting sea ice after the snow has melted. The SSL is a porous, granular, and highly fragile pillared structure on top of the ice, which effectively backscatters solar radiation and keeps the surface albedo of melting ice relatively high (Macfarlane et al., 2023). Due to the small-scale characteristics of the SSL, it is difficult to relate the surface albedo of the SSL to the available variables of a climate model with spatial scales in the order of several kilometers. Consequently, the surface albedo of the SSL is a critical issue in the albedo parametrization. Since the albedo of bare ice is generally lower than the albedo of the SSL, the surplus of radiation energy at the ice surface may lead to an amplified melting of sea ice in the model.

Simulations and ground-based measurements of the seasonal evolution of surface albedo during MOSAiC were previously presented by Light et al. (2022). The authors used an Earth system model (1° spatial resolution) for comparison with surface albedo measurements manually made along three survey lines. These measurements could not be performed with the same high temporal frequency during the complete campaign for logistical reasons. Therefore, the transition from dry to wet snow during the onset of melting was less captured than in our study, which relied on autonomous measurements from a radiation station. Similar to our results, Light et al. (2022) showed that in particular the representation of melt pond albedo in the model needs to be improved, while the general surface albedo values and properties of the different ice types were captured well.

We investigated how the surface albedo model bias affects the balance between incoming and outgoing irradiance at the surface by calculating the net solar irradiance. The direct comparison of model results and aircraft observations yielded a RMSE of $30.2\,\mathrm{W\,m^{-2}}$. This error can be primarily attributed to differences in surface albedo. However, the ranking of the standardized

regression coefficient suggests coefficient suggests that uncertainties in the modeled cloud cover also contribute to the model
bias in net irradiance. The direct effect of the surface albedo bias on net irradiance was derived from offline evaluation against
different airborne measurement data. We found a smaller spread between modeled and parameterized net irradiance (RMSE =
13.5 W m$^{-2}$) compared to the HIRHAM-NAOSIM run. This improvement is partly due to the fact that the cloud cover, which
influences the parametrization of the surface albedo, was derived from the measurements and not from the model as the model
produces too many cloudy cases (see above). The impact of the surface albedo bias on the net irradiance as a function of the
cloud optical depth revealed a significant negative bias (median: -6.4 W m$^{-2}$) for optically thin clouds (equivalent COD < 5),
while for optically thicker clouds (equivalent COD > 5) a median bias value of only 0.1 W m$^{-2}$ was determined.

From this analysis, it appears that a change in the surface albedo scheme based on temporally limited measurements requires
an assessment for other time periods and regions with different atmospheric and sea ice conditions. Weaknesses in the surface
albedo scheme have seasonally varying effects, as exemplified for HIRHAM-NAOSIM. Uncertainties of the surface albedo
dependence on clouds especially affect the surface albedo in spring, whereas in the melting season mainly the surface type
parametrization determines the accuracy of the surface albedo scheme. We invite the modeling community to use this air-
borne data set to evaluate other surface albedo schemes, as it provides decoupling of surface type fraction and surface albedo
parametrization for larger spatial scales than covered by ground-based observations. This is advantageous because an incorrect
type fraction can be compensated by an incorrect specific albedo of the surface type, which then leads to an apparently con-
sistent total surface albedo (Light et al., 2022). However, in order to further improve the existing parametrizations, the local
ground-based observations, especially from MOASiC, will be crucial in describing the surface-type specific dependencies, as
most of the potential parameters influencing the surface albedo were directly measured during MOSAiC.

*Data availability.* Pyranometer and KT-19 data are published on PANGAEA (https://doi.org/10.1594/PANGAEA.900442; Stapf et al. (2019),
https://doi.pangaea.de/10.1594/PANGAEA.932020; Stapf et al. (2021), https://doi.pangaea.de/10.1594/PANGAEA.936232; Becker et al.
(2021)). A joint surface albedo and surface type fraction data set can be downloaded from PANGAEA (https://doi.pangaea.de/10.1594/
PANGAEA.963001; Jäkel et al. (2023a), https://doi.pangaea.de/10.1594/PANGAEA.963064; Jäkel et al. (2023b), https://doi.pangaea.de/10.
1594/PANGAEA.963078; Jäkel et al. (2023c), https://doi.pangaea.de/10.1594/PANGAEA.963106; Jäkel et al. (2023d), https://doi.pangaea.
de/10.1594/PANGAEA.962996; Jäkel et al. (2023e)). The MOSAiC radiation stations data are available on Pangaea (https://doi.pangaea.
de/10.1594/PANGAEA.949556). The processed MPD albedo product is available from https://data.seaice.uni-bremen.de/databrowser/#p=
MERIS_OLCI_albedo. HIRHAM–NAOSIM data are available at the tape archive of the German Climate Computing Center (DKRZ; https:
//www.dkrz.de/en/systems/datenarchiv). We will also make subsets of the data available via Swift (https://www.dkrz.de/up/systems/swift) on
request. AMSR sea ice concentration data were obtained from National Snow & Ice Data Center (https://nsidc.org/data/AU/_SI6/versions/1,
Meier et al. (2018))

## Appendix A:  Radiative transfer simulations

To calculate the solar broadband upward and downward irradiance the radiative transfer package libRadtran (Mayer and Kylling, 2005; Emde et al., 2016) was applied. The Discrete Ordinate Radiative Transfer solver (DISORT; Stamnes et al., 2000) was used with pseudo-spherical geometry to account for the low sun conditions in the Arctic. The absorption parametrization after Gasteiger et al. (2014) and the extraterrestrial spectrum was taken from Gueymard (2004) were chosen. The atmospheric standard profiles of trace gases, temperature, pressure, and humidity for Arctic summer and winter, respectively, were adjusted

to measurement conditions using radio sounding data from Ny-Ålesund (Maturilli, 2020).

    In libRadtran clouds can be defined by water content and effective radius at each model layer. For each flight day under cloudy conditions a standard profile was created as listed in Tab. A1.

**Table A1.** Model setup of clouds for libRadtran.

| Day and Campaign | Conditions | Cloud base and cloud top | Reference |
|---|---|---|---|
| 25 March 2018 (PAMARCMiP) | cloudless | | |
| 29 March 2022 (HALO-$(\mathcal{AC})^3$) | cloudless | | |
| 23 March 2019 (AFLUX) | cloudy (mixed phase) | 90 - 540 m | Moser and Voigt (2022a) |
| 3 April 2018 (PAMARCMiP) | cloudy | 400 - 600 m | NA |
| 4 June 2017 (ACLOUD) | cloudy | 100 - 350 m | Chechin (2019) |
| 25 June 2017 (ACLOUD) | cloudless | | |
| 13 September 2020 (MOSAiC-ACA) | cloudy | 340 - 460 m | Moser et al. (2022b) |

*Author contributions.*  EJ, MW, and WD designed this study. EJ wrote the main text and prepared the figures. EJ, SB, HN, RT, JB, and MN analyzed the observational data sets and contributed to the interpretation of the results. WD performed the model simulations. AR and WD

provided guidance to apply and describe the albedo scheme of HIRHAM–NAOSIM. All co-authors helped with paper edits.

*Competing interests.*  The authors declare that no competing interests are present.

*Acknowledgements.*  We gratefully acknowledge the funding by the Deutsche Forschungsgemeinschaft (DFG, German Research Foundation) – Project Number 268020496 – TRR 172, within the Transregional Collaborative Research Center "ArctiC Amplification: Climate Relevant Atmospheric and SurfaCe Processes, and Feedback Mechanisms (AC)3". Further, the authors would like to thank the Federal Ministry of

Education and Research (BMBF) for funding the project ALIBABA under grant 03F0870. WD and AR acknowledge the funding by the European Union's Horizon 2020 research and innovation framework programme under Grant agreement no.101003590 (PolarRES project). Data used in this manuscript were produced as part of the international Multidisciplinary drifting Observatory for the Study of the Arctic

Climate (MOSAiC) with the tag MOSAiC20192020 and the Project_ID: AWI_PS122_00. We thank Manuel Moser from DLR for providing and discussing the in situ cloud property data.

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
