# Peer review of "Observations and modeling of areal surface albedo and surface types in the Arctic"

_EGUsphere, 2023_

## Author Comment (AC1)

**Reply to Reviewer #1:**

**We thank David Bailey for the time and efforts he spent reading our manuscript and providing valuable advices. Please find below a discussion of the reviewer's comments (italic). Changes/additions made to the text are underlined and given in quotes.**

*This is a very nice study that assesses surface albedo from the MOASAiC campaign versus the HIRHAM-NAOSIM model. Maybe a lot of it is my misunderstanding of what was done here. The pieces are mostly here and I do think this is a worthwhile study, but I have some fairly significant concerns here.*

*1. In terms of originality, Light et al. have recently published a similar study in Elementa. While there is more emphasis on the model results here, I still think some additional contrast to what they found would be useful here.*

Following the Reviewer's comment we refer to this study in the Section 5 when summarizing the results of the comparison with MOSAiC data:

"Simulations and ground-based measurements of the seasonal evolution of surface albedo during MOSAiC were previously presented by \cite{Light_2022}. The authors used an Earth system model (1° spatial resolution) for comparison with surface albedo measurements manually made along three survey lines. These measurements could not be performed with the same high temporal frequency during the complete campaign for logistical reasons. Therefore, the transition from dry to wet snow during the onset of melting was less well observed than in our study, which relied on autonomous measurements from a radiation station. Similar to our results, \cite{Light_2022} showed that in particular the representation of melt pond albedo in the model needs to be improved, while the general surface albedo values and properties of the different ice types were captured quite well."

In the context of the study by Light et al., we did not explicitly mention the difficulty of evaluating the modeled surface type fraction from ground-based observations. Even using observations along a 200 m survey line instead of single point measurements will hardly represent the variability within a modeled grid cell with 1° spatial resolution. At the end of Section 5, we write:

"We invite the modeling community to use this airborne data set to evaluate other surface albedo schemes, as it provides decoupling of surface type fraction and surface albedo parametrization for larger spatial scales than covered by ground-based observations."

*2. My biggest concern is the bias in absorbed shortwave (irradiance), Figure 8. Did the authors compare the incoming shortwave between the model and observations? The albedo could be perfectly correct, but if the incoming shortwave is biased, then the absorbed will be similarly biased. I am not an expert in atmospheric radiation, but I think it would be helpful to see a comparison of incoming and outgoing shortwave. Perhaps this was mentioned, but I think this could be expanded upon.*

Indeed, the net irradiance is highly dependent from the representation of the incoming (here called downward) irradiance. An underestimated modeled downward irradiance directly leads to an underestimation of the modeled net irradiance assuming the same surface albedo. Therefore, we included the information on the bias of modeled downward irradiance in Figure 8. The size of the symbols directly corresponds to the difference of the modeled and measured downward irradiance. To make this clear, we adjusted the figure caption:

[Figure]

**Figure 8.** Scatterplot of net irradiance based on measured and modeled surface albedo covering the flights performed during PAMARCMiP and MOSAiC-ACA. The horizontal bars indicate the standard deviation of the averaged measured $F_{net}$. Color code gives the surface albedo difference ($\Delta\alpha = \alpha_{model} - \alpha_{meas}$) and the symbol size the difference of the modeled and measured downward irradiance.

[Figure]

Not included in the manuscript: a figure showing the measured and modeled downward irradiance. The mostly negative bias of the modeled F↓ is visible. In the manuscript, we report on the mean deviation.

We rephrased parts of this section:
"The $F_{net}$-differences between measurement and model depend not only on $\Delta\alpha$, but we must also take into account the difference in the downward irradiance ($\Delta F\downarrow$). A negative $\Delta F\downarrow$ (smaller symbols in Fig. 8) may occur when the modeled extinction of F↓ caused by modeled clouds is higher than an observation would show. This is especially the case when cloudless situations were observed but not modeled. It would lead to an underestimation of the modeled net irradiance, assuming the same surface albedo. In fact, a mean negative bias of the modeled F↓ (mean $\Delta F\downarrow$ = -31 W m−2) was found, which can be related to an overestimation of the modeled cloud cover. However, the downward irradiance itself also depends on the surface albedo. In particular, below clouds F↓ is enhanced for brighter surfaces due to multiple-scattering between surface and cloud base. A positive surface albedo bias would lead to a positive bias in F↓, assuming a similar cloud representation. On average $\Delta\alpha$ was 0, indicating a small effect of surface albedo on the modeled F↓.
Overall, both cloud properties and surface albedo must be well represented for modeling net irradiance correctly. To estimate whether the representation of clouds or the surface albedo potentially contribute more to the uncertainty of $F_{net}$, we calculated …"

*3. On a similar note, the authors talk about the importance of albedo for climate model simulations. However, related to point 2, we often have to adjust the snow albedo to compensate for biases in the incoming shortwave. So, it is possible to have the "correct" albedo, but for the wrong reasons.*

The surface albedo is a crucial parameter for modeling radiative transfer in the atmosphere, especially for calculating the upward irradiance. The downward irradiance at the surface is very sensitive to the properties of the atmospheric components (aerosol and cloud particles, trace gases, ...). Therefore, in atmospheric applications, these components must be properly reproduced by the model to obtain a correct model output. Adjusting the surface albedo to get a correct downward irradiance is less effective. However, if you are interested in modeling radiative transfer within a snow layer, for example, you need the incident irradiance, which can be adjusted by changing the

surface albedo. Perhaps the reviewer is aiming in this direction. Since we are interested here in the atmospheric solar irradiance effect of surface albedo, we would rather suggest that the atmospheric parameters be adjusted so that the downward irradiance is well reproduced by the model.

*4. What is the temporal resolution here? It wasn't obvious to be if these are instantaneous, hourly, etc. I assume the model is saving the fields at the same temporal resolution? How is albedo defined when there is no sun?*

The temporal resolution of the model output for MOSAiC (2020) was three hours, and one hour for PAMARCMiP (2018). It is mentioned in Section 2.3:

"The model output was given with a spatial resolution of about 27 km distributed over 200 x 218 grid points on a circum-Arctic domain. [...] The HIRHAM-NAOSIM model was run for 2018 covering the time frame of the PAMARCMiP campaign (temporal resolution of 1 hour), and for the entire MOSAiC period (temporal resolution of 3 hours) that includes the time frame of the ground-based measurements from spring to autumn 2020 and the period of the aircraft observations during MOSAiC-ACA."

For the comparison of measured and model data, the spatial and temporal overlap between the two data sets was taken into account when filtering the data.

*How is albedo defined when there is no sun?*

Since the surface albedo is not determined by a dependence of the solar zenith angle, there is no difference for the case when there is no sun. All data considered in this study were taken during the presence of the sun (polar day).

*5. I'm very confused about the use of "online" and "offline" models here. Is the difference that one has prognostic radiation and the other has specified radiation? I would like the authors to expand upon the description of these. I think this is where you are trying to get at the question raised earlier about whether the incoming shortwave is biased, or the albedo is biased. I think a bit more could added to section 4.2 to help alleviate these concerns.*

We have tried to explain the differences between online and offline simulations in section 2.4. The "offline" mode applies only the two parameterizations of subtype albedo and subtype fraction as they are implemented in HIRHAM-NAOSIM. It uses measured parameters that were derived from the observations along the flight tracks. In contrast, for the "online" simulations, the HIRHAM-NAOSIM model package was run completely independently of the measurements. So, the results discussed in Section 4.1 refer to an "online" application of the model, whereas Section 4.2 takes only the parametrizations into account.
We have adapted the beginning of Section 4.2 to introduce the independent radiative transfer simulations that allow a sensitivity study of the $F_{net}$ dependence on surface albedo.

"In contrast to the study of the HIRHAM-NAOSIM results, the application of the offline evaluation allows a reduction in the dependencies of the Fnet bias for the comparison of the parameterization with the airborne measurements. The measured subtype fractions were used to identify only the influence of the bias of the parameterized surface albedo on Fnet, without having to consider the uncertainties of the subtype fraction parametrization. The net irradiance was determined along the flight path for seven selected days during all five flight campaigns, covering cloudy and cloudless conditions. Radiative transfer simulations were performed for these cases using the measured and parameterized surface albedo. In this way, the sensitivity of net irradiance to surface albedo was

quantified under the same predefined atmospheric condition. These conditions were matched to the measurements made during the selected flights (see Appendix A).”

*Minor points.*

*1. In figure 3, the panels that show the surface type are hard to see (a, g, c, i). Maybe just lines instead of filled contours. The red of melt ponds in particular is hard to see.*

It is true that the proportions of each type are difficult to read when their contributions are small, as in the case of melt ponds. However, we deliberately chose to use a stack plot so that we could immediately identify the dominant surface types. Individual lines, as suggested by the reviewer, would not be helpful because the temporal variation is quite high.
The proportions of surface types are presented as a stacked area plot to identify the predominant subtypes. We have improved the figure caption to better indicate the surface types, and added the following to the figure caption:

“Figure 3. (a) - (j) Temporal development of surface types, surface albedo (blue lines; left y-axis) and surface skin temperature (grey lines; right y-axis) for all five flight campaigns. The proportions of surface types are presented as a stacked area plot to identify the predominant subtypes. Vertical green lines separate the individual flight days. Dates given in the panels are explicitly mentioned in the text. (k) Averaged surface albedo as a function of sea ice fraction (bin size of 10 %), separately for cloudless and cloudy conditions. The standard deviation of the averages is represented by thin vertical bars.”

At the beginning of Section 3.1, we mentioned the type of plot directly:

“An overview of the proportions of classified subtypes along the flight tracks of the five campaigns is shown in Fig. 3 as a stacked area plot. The temporal development …”

*2. In figure 4, I prefer you not use the description of "violin" plot. While this might describe the shape it doesn't say anything about what you are showing. Just a description of what you are showing is sufficient. Also, you could refine the Y-axis. Everything below 0.6 is not interesting in spring and summer.*

We have replaced the term “violin” plot by “distribution” in the figure caption. A deeper description of the figure is given in the main text.

**Figure 4.** Distributions of measured and modeled surface albedo separated in cloudy (red distribution) and cloudless (blue distribution) cases for the seasons spring, summer, and autumn. The median value (also indicated by the white line) is given together with the first and third quartiles (black dashed lines).

We prefer to keep the y-axis in order to have a uniform scale for all seasons. This facilitates comparability between the individual distributions.

*3. Similarly in Figure 5c. Are you simply reflecting the same information on both sides of the line?*

The four individual distributions are symmetrical because they are not divided into cloudy and non-cloudy cases as in Fig. 4. Similar to Figure 4, we have changed the figure caption:

(same color code as in (a)). (c) Distributions of satellite, model, and aircraft surface albedo data. Left: statistics of model and MPD retrieval for the area that is covered by the satellite data shown in b), right: comparison along the flight path only. (d) Time series of modeled mean

*4. Figure 7b is a similar issue to point 1. I find that these "stacked" plots are kind of tricky to interpret. Maybe line plots are better here.*

For consistency with Figure 3, we would like to keep the stacked area plots.

---

## Author Comment (AC2)

**Reply to Reviewer #2:**

**We thank the reviewer for the time and efforts she/he spent reading our manuscript and providing valuable comments. Please find below a discussion of the reviewer's comments (italic). Changes/additions made to the text are underlined and given in quotes.**

*General comment*

*I this paper, the Arctic surface albedo simulated with the coupled regional climate model HIRHAM-NAOSIM is evaluated with aircraft and surface-based observations collected during several field campaigns. The study is very relevant for the polar modelling community, the applied method is convincing, and the observational dataset used for the model validation is outstanding. However, I have few major concerns:*

1. *The text in Sect 3 and 4 is hardly readable, the expressions are unclear, the language is not suitable for scientific publication and needs to be extensively rewritten. In my detailed comments I only point to few examples, but almost all the sentences require improvement.*

   We have revised the text considerably according to the reviewer's comment, and have rephrased the relevant text passages. We hope that this improves the readability. For details please refer to the detailed comments below.

2. *In some cases, the interpretation of the results needs to be deepened (see my detailed comments). Some results depend on the selected regions and time of the year and cannot be generalized (such as the relative impact of clouds or albedo biases on the bias in surface net irradiance).*

   Please read our responses to the detailed comments below. They refer to that general remark.

3. *In my view, one of the most striking results is the model underestimation of surface albedo after the onset of melting (Fig 7). The underestimation is explained as due to the fact that, when snow disappears, the ice surface is represented as bare ice and not as the surface scattering layer that forms during the melting. This result deserves more discussion.*

   We followed the Reviewer's suggestion and added a discussion of the issue associated with the missing surface scattering layer, and we included the reference to Macfarlane et al. (2023). For details, please refer to our response to the corresponding detailed comment.

*Detailed comments:*

*Abstract: the result related to the lack of proper representation of the surface scattering layer over melting sea ice is missing from the abstract. I believe it is relevant to include it.*

The reviewer points out an important issue, namely the need for improvements in the albedo parameterizations to account for an SSL. Although we have not quantified the impact of SSL on surface albedo, we have observed an underestimation of modeled surface albedo when only bare ice is considered in the model after snowmelt. We added the following statement in the abstract:

"The lack of an adequate model representation of the surface scattering layer formed on bare ice contributed to the underestimation of surface albedo in summer."

*line 176: "…where sea ice is further divided into snow-covered ice (subscript s), bare ice (subscript bi), and melt ponds (subscript mp)": could you please add a comment on which ice category the "surface scattering layer (SSL)" (also called "white ice") belongs to? It is not snow but very much resembles it, being much more reflective than bare ice (for the definition of SSL see e.g. https://online.ucpress.edu/elementa/article/11/1/00103/195863/Evolution-of-the-microstructure-and-reflectance-of and https://agupubs.onlinelibrary.wiley.com/doi/full/10.1029/2006JC003977).*

*From your Table 3, the surface scattering layer would belong to "snow-covered ice" category when looking at the albedo intervals.*

The evolution of the surface scattering layer (SSL) on bare ice is not considered in HIRHAM-NAOSIM. We agree that the SSL is an important component for albedo parameterization. However, it has not yet been accounted for in HIRHAM-NAOSIM by a separate class. For the classification of surface types from camera observations, we combined SSL/white ice into a common class "snow-covered ice/white ice" due to similar reflectance properties.

We added the following:

"Note that the surface type "white ice", which results from a highly reflective scattering layer on top of melting ice (Macfarlane et al., 2023), is not explicitly considered in HIRHAM-NAOSIM. Due to its higher albedo compared to bare ice, white ice is added to the class of snow-covered ice in this work and classified accordingly based on camera observations during the measurement flights."

*Fig 4: very nice Figure!!*

Thanks.

*Sect 3 and 4: reading these sections is extremely painful because of the unclear text and imprecise vocabulary. The logical rigor of the sentences is poor as there are often missing logical steps in the explanations. The text is not suitable for scientific publication and needs to be extensively rewritten. I provide here only some examples of poor sentences*

We thank the reviewer for her/his detailed suggestions to improve the content and language.

*line 300-301: "From that, we assume that the distribution shown for the modeled surface albedo is biased to higher values, since the cloud cover is overestimated." This is a quite badly expressed sentence and concept. Maybe you mean something like "Based on these results, we argue that the match between satellite- and model-derived surface albedo medians results from the compensation of two opposite model biases: the overestimation of modelled clouds, which caused a positive bias in modelled albedo, was compensated by a negative bias in modelled clear-sky albedo." Do you agree?*

Yes, and you have understood what we were trying to say. We have improved and revised this text considerably to make the text clearer. The adjusted text reads:

"As the modeled surface albedo is cloud cover dependent, the representation of the clouds in the model must be taken into account to evaluate the modeled surface albedo. While the aircraft and satellite observations showed mostly cloudless conditions, the model calculated a cloud cover of about 100 % in most areas. Based on these results, it can be assumed that the match between satellite- and model-derived surface albedo medians results from the compensation of two opposite model biases: the overestimation of modeled cloud coverage, which caused a positive bias in modeled surface albedo, was compensated by a negative bias in modeled cloudless surface albedo."

*lines 315-316. "At end of March, a distinct minimum of sea ice coverage (0.86) was simulated for the area covered by the flight on 3 April 2018 leading directly to the minimum of the surface albedo." Totally unclear sentence, I did not manage to guess what you mean.*

We have revised this text to clarify what is meant. It reads now as follows:

"The temporal variation of the modeled surface albedo is illustrated in Fig. 5d. Each individual line represents the time series of the area-averaged surface albedo for one of the seven overflown areas. In addition, the mean measured surface albedo (including standard deviation) on the corresponding day is shown. […] The albedo time series of the area overflown on 3 April 2018, shows a pronounced albedo minimum in late March associated with a minimum sea ice cover (0.86)."

*line 316-318: "The corresponding measured areal-averaged surface albedo shows, on the one hand, a much greater spatial variability and, on the other hand, a clear tendency towards smaller surface albedo values. This tendency…" Please rephrase, and not use the word "tendency" if you are not showing a decreasing/increasing trend in your time series, it is very misleading. Do you mean that area-averaged modeled albedo is positively biased compared to area-averaged aircraft observations? If so, write it clearly.*

We rephrased that part as follows:

"In general, the measured surface albedo shows much greater spatial variability but smaller averaged values than the model. This positive bias of modeled surface albedo cannot be explained by a lower sea ice coverage modeled with HIRHAM-NAOSIM."

*lines 332-333: "This partly explains the difference in the distribution of modeled and measured surface albedo, in particular for the surveyed regions on September 8 and 13." You did not show this result: either you show the plot, or you remove this sentence.*

Perhaps the word "distribution" is misleading here. From the colors in Fig. 6a, one can conclude that the modeled albedo for these two days has a negative bias. The average values are shown in Fig. 6b, which support the statement.

"This could partly explain the difference between the modeled and measured surface albedo. The modeled albedo map shows a negative bias compared to the measurements along the flight path (overlaying brighter points in Fig. 6a), especially for the flights on 8 and 13 September."

*line 336: "Cloud effects are small, as mostly a full cloud coverage was modeled by HIRHAM-NAOSIM." This sentence cannot be understood if you don't include all the logical steps. Do you mean that "Clouds did not significantly affect the temporal variability of modeled and observed surface albedo because both observations and model simulations were carried out in overcast conditions"?*

In general, greater variability in modeled surface albedo may also be caused by variable cloud conditions, since the albedo parameterization distinguishes between cloudy and cloudless conditions. For the MOSAiC-ACA cases presented here, however, both the model and the measurements show predominantly cloudy conditions. That's why we argue that clouds do not contribute to the modeled variability here.

We rephrased the sentence:

"Since HIRHAM-NAOSIM mostly simulated a cloud coverage of 100 %, the variability of the surface albedo cannot be due to the use of different parametrizations for cloudy and cloudless conditions."

*line 337-338: "The measured areal-averaged surface albedo shows best agreement for the region overflown on September 2, although parts of the northernmost section of this flight path were underestimated by the model." It seems to me that part of the northernmost section of that flight is overestimated by the model (the western part) and part is underestimated (the eastern part).*

Indeed, the flight section following mostly the same latitude (northernmost section) is showing partly an overestimation of the modeled albedo. We changed it:

"The measured area-averaged surface albedo shows best agreement for the region overflown on September 2, although the surface albedo along the northernmost section of this flight path was partly overestimated by the model."

*Sect 3.3.3: it would be good to explain why you decided to use the data from one RAMSES spectral albedo station and not from other albedo stations (there were several broadband albedo stations and other RAMSES stations), especially because for this study the spectral to broadband albedo conversion. Wouldn't be more straightforward to apply broadband observations? How representative of the MOSAiC ice floe surface the data collected at the selected station are?*

We agree, that all other data sets of albedo measurements made during MOSAiC have great potential. However, we are sticking with the RAMSES data, mainly because of its continuity. The 2020R12-RAMSES station was installed during leg 3 at the L3 distributed network site. It was operated between April 24, 2020 and August 07, 2020. We selected the RAMSES station because it is independent of the logistic gap during MOSAiC, where observations had to be interrupted. Since the time frame of this logistic gap exactly covers the transition from dry to wet snow during the onset of melting, we used the autonomous measurements for comparison with HIRHAM-NAOSIM to have a continuous data set available.

We are aware that the data from the surface flux stations also provide almost continuous time series of irradiance measurements during MOSAiC. The data sets have already been used in another publication by Foth et al. (2023), which is currently under discussion in TC. In this work, the HIRHAM-NAOSIM model was evaluated against the flux station data, focusing on the changes in the snow albedo parameterization with respect to clouds (presented in Jäkel et al., 2019). To avoid repetition, we did not use this data set in our study. We refer to this study in Sec. 1:

"A comparison of the modeled surface albedo between the revised model and the earlier version was presented by Foth et al. (2023). They evaluated both model versions using measurements from two flux stations deployed during MOSAiC. They found that the revised snow surface albedo parameterization led to a more realistic simulation of surface albedo variability during the snowmelt period in late May and June."

Foth, L., Dorn, W., Rinke, A., Jäkel, E., and Niehaus, H.: On the importance to consider the cloud dependence in parameterizing the albedo of snow on sea ice, EGUsphere [preprint], https://doi.org/10.5194/egusphere-2023-634, 2023.

And in Sec. 3.3.3 we added:

"During MOSAiC, the seasonal evolution of the surface albedo was measured by autonomous radiometers. In this study, data from one of the RAMSES stations (2020R12, following the notation of Tao et al., 2023) were used. 2020R12 was deployed on second year ice at site L3 of the MOSAiC Distributed Network (Nicolaus et al., 2022). This data set provides almost continuous time series of irradiance measurements between 24 April and 7 August 2020, which in particular allow the observation of the

transition from dry to wet snow during the onset of melting. We applied the two corrections according to Eqs. (1) and (2) to the ground-based observation of the autonomous radiometers…"

As requested by the other reviewer, we have referred to the work of Light et al. (2022) and mentioned the role of autonomous measurements:

"Simulations and ground-based measurements of the seasonal evolution of surface albedo during MOSAiC were previously presented by \cite{Light_2022}. The authors used an Earth system model (1° spatial resolution) for comparison with surface albedo measurements manually made along three survey lines. These measurements could not be performed with the same high temporal frequency during the complete campaign for logistical reasons. Therefore, the transition from dry to wet snow during the onset of melting was less well observed than in our study, which relied on autonomous measurements from a radiation station."

*line 359: the reference (Tao et al., 2023) is missing.*

The manuscript by Tao et al. "Seasonality of spectral radiative fluxes and optical properties of Arctic sea ice during the spring-summer transition" is still in review (in Elementa). We will remove the citation in case the manuscript is not accepted in time.

Tao, R., Nicolaus, M., Katlein, C., Anhaus, P., Hoppmann, M., Spreen, G., Niehaus, H., Jäkel, E., Wendisch, W., and Haas, C.: Seasonality of spectral radiative fluxes and optical properties of Arctic sea ice during the spring-summer transition, submitted to Elementa: Science of the Anthropocene, 2023.

*line 368-370: "On June 21 and June 22, both data sets showed a similar surface albedo, even though the spatial variation of the satellite product was smaller than the temporal variability of the ground-based surface albedo measurement within a day." In my opinion, comparing spatial variability of one data with temporal variability of another data is not meaningful. You need to better elaborate, otherwise the comparison between in situ and satellite data is meaningless.*

Actually, we want to point out that the spatial variability of the observed surface albedo within the footprint of the HIRHAM grid at a fixed time can be smaller than the variability of the surface albedo on a single day. However, since this statement may be deficient, because small-scale changes cannot be resolved by the satellite observations, we have adjusted this sentence:

"On June 21 and 22, satellite and ground-based measurements showed a similar mean surface albedo of 0.69. For the observed cloudless conditions, Eq. (1) can be applied to correct the radiometer measurements."

*line 391, eq. 10: the way in which the equation is presented is misleading: the difference between measured and modelled net irradiance does not depend on albedo alone but also on the incoming irradiance. Please correct.*

We agree and generalized Eq. 10 as follows:

$$\Delta F_{\text{net}} = F_{\text{net,model}} - F_{\text{net,meas}} \quad . \tag{10}$$

*Sect 4.1, lines 406-417: the authors did a linear regression analysis to assess the relative impact of biases in albedo, solar zenith angle and modelled cloud water path on the bias in modelled net shortwave irradiance. I think that the results are very much dependent on the considered dataset*

*(March-April and September observations). A different dateset, with different spatial and temporal variability in albedo and cloud properties would provide different standard deviations with respect to model simulations, yielding a completely different result. For instance, I would expect that in summer, when albedo is lowest, cloud optical thickness is largest, and shortwave cloud radiative forcing is largest (most negative), the cloud std may cause a larger error in surface net shortwave irradiance than the std in albedo. Hence, I recommend considering the results from the perspective of the analyzed dataset and discuss the implication of different albedo and clouds conditions.*

Yes, the reviewer is right, that the coefficients we derived in this study cannot be generalized and depend on the specific conditions of the selected data set. Therefore, we wrote: **"For the analyzed cases during MOSAiC-ACA and PAMARCMiP** we found the strongest impact of the surface albedo ($\beta_\alpha$ = -0.80), and less impact of the CWP ($\beta_{CWP}$ = -0.38) and SZA ($\beta_{SZA}$ = -0.23)."

Inspired by the reviewer, we have calculated the monthly beta-parameters for a specific area (flight area from September 2) as shown in the plot below. The three lowest panels depict the monthly means and monthly standard deviations of the surface albedo, CWP, and SZA. They support the reviewer's considerations that the beta parameters are variable over time and are associated with a reduction in albedo and an increase in CWP in summer. We further find that the magnitude of $\beta_{SZA}$ in this time series is higher throughout the year than for the two time periods (April/March and September) we considered in the study. This is mainly due to the fact that we included complete daily cycles in the evaluation (all data points with SZA < 85°), whereas the data selection of PAMARCMiP and MOSAiC-ACA was limited to the times of the measurement flights. Since this is only a quick analysis, we will not make any quantified statements about potential changes of the beta parameters here. This could be the subject of a new study, but is beyond the scope of the current paper. Here, we have added the following:

"However, we expect a seasonal dependence of the standardized regression coefficients. According to Eq. (11), a stronger variability of the individual parameters contributes to a higher magnitude of $\beta_j$. In summer, clouds tend to have a high cloud water path with a high variability, while the surface albedo reaches its minimum. Therefore, it is assumed that the contribution of the surface albedo bias to the $F_{net}$ uncertainty is reduced, whereas the model representation of cloud properties gets more relevant compared to the two periods shown in this study ."

[Figure]

*Sect 4.2 and related text in Sect 5: often the expressions "surface albedo forcing" or "surface albedo effect" on the net shortwave irradiance are improperly used, as in reality you meant "impact of the surface albedo bias on the calculated net shortwave irradiance". This is very confusing. I recommend rewriting the text paying particular attention to the precision of the used vocabulary and espressions.*

By analogy with cloud radiative forcing, which is defined as the difference between the net irradiance under cloudy and cloudless conditions (Ramanathan et al., 1989), we have used the term surface albedo forcing because it indicates the difference between the net irradiance derived for the modified condition (parameterized albedo) and the reference condition (measured albedo). However, we agree that this expression could be confusing, and therefore, we now omit this term as suggested by the reviewer. The text was changed accordingly:

"The maximum impact of the albedo bias on ΔFnet is derived for cloudless summer conditions (ΔFnet=±80 W m-2). For the same range of Δα in spring, ΔFnet is found to be less than half of its magnitude in summer (ΔFnet=±35 W m-2)."

"We investigated how the surface albedo model bias affects the balance between incoming and outgoing irradiance at the surface by calculating the net solar irradiance."

Ramanathan V. et al.: Cloud-Radiative Forcing and Climate: Results from the Earth Radiation Budget Experiment, Science 243, 57-63, DOI:10.1126/science.243.4887.57, 1989.

*lines 445-446: "This indicates that a surface albedo bias in spring is less relevant for the absolute amount of the solar energy budget at surface than in summer.", and 495-497: "Since the maximum surface albedo effect on the net irradiance was derived for cloudless summer conditions, it can be concluded that the surface albedo bias is more relevant to the absolute amount to the solar energy budget in summer than in spring." Even if the surface albedo bias causes a larger bias in clear-sky surface net irradiance when the incoming irradiance il largest (in summer), it does not mean that it is more relevant for the summer than for the spring surface energy budget. In fact, cloudless skies are much more frequent in spring than in summer. However, the freezing temperatures make the albedo spatial and temporal variability and, thus, the bias in modelled albedo, much smaller in spring than in summer. I wish the authors could expand the discussion on this result, including references to previous studies.*

Thanks for bringing this up. In contrast to the statement of the reviewer ("… and, thus, the bias in modelled albedo, much smaller in spring than in summer. ") we observed an increased albedo bias with a wider distribution in spring than in summer (see figure below). This may not be true for other models where the cloud dependence of the surface albedo parameterization is not considered. In our study, however, we see an overestimation of the modeled surface albedo, as the cloud dependence in the albedo parameterization is deficient for optically thin clouds, which contributes to the albedo bias in spring.

[Figure]

We rephrased the text in Sec. 4.2 and removed the sentence in Sec. 5 to weaken the statement that the surface albedo bias is more relevant in summer:

"For the same range of Δα in spring, ΔFnet is found to be less than half of its magnitude in summer (ΔFnet=±35 W m-2). In spring, however, we observed from the flight measurements an increased albedo bias with a wider distribution (Δα=0.02±0.07) than in summer (Δα=0.00±0.04 ). This means

that greater effects on the solar radiation balance between solar incoming and outgoing irradiance due to the surface albedo model bias can be observed in summer, but these are less likely than in spring."

The reason for the poor representation of the surface albedo by the model in spring was investigated in more detail with the help of Fig. 9c.

*lines 472-473: "We conclude that a functional dependence, rather than a pure discrimination between cloudy and cloudless conditions, is required to properly describe the cloud effect on surface albedo." The advocated physical dependence of the broadband albedo parameterization on cloud properties (optical thickness) is much less physically consistent than the waveband-dependent albedo parameterization would be. Only a waveband-dependent albedo parameterization that at least distinguishes between visible and infrared regions can account for the cloud impact on albedo in a manner that retains the coupling and dependencies between the physical variables. Could this solution be applied in HIRHAM-NAOSIM? I invite the authors to consider this solution or at least to comment on it.*

We completely agree that the use of a waveband-dependent albedo parameterization would be physically more consistent. However, the implementation of such a complex parameterization is very time-consuming and much more complicated than introducing a simple cloud dependence in the present broadband albedo parameterization. Furthermore, and indeed the simulation results indicate that the cloud effect on surface albedo can be reasonably reproduced using a broadband albedo parameterization with cloud dependence. Based on our results we argue that the problems in the present albedo parameterization are rather related to deficiencies in the simulation of clouds and the surface fractions than to poorly reproduced cloud effects. These deficiencies will not be remedied by switching to a waveband-dependent albedo parameterization. Therefore, we are convinced that our results using a simpler broadband albedo parameterization are helpful and indicate that we should focus on improving the representation of clouds and surface type fractions in the models next, before implementing the more complex albedo parameterization. Basically, we agree that a waveband-dependent albedo parameterization should be implemented on a longer perspective.

We added:

"In the absence of a waveband-dependent albedo parameterization, the consideration of a simple cloud dependence in the broadband albedo parameterization is able to mimic the cloud effect on surface albedo reasonably. The cloud effect might be further improved by a more sophisticated functional dependence on cloud cover or cloud water content, rather than a pure distinction between cloudy and cloudless conditions."

*lines 488-489: "In particular, the surface albedo of the scattering layer classified as bare ice seemed to be underestimated." This is a critical issue: currently, in all sea ice schemes that I am aware of, the albedo of ice without snow is simulated as the albedo of bare ice, which is much lower than the albedo of the surface scattering layer. The surface scattering layer is completely ignored in the sea ice surface schemes, with the consequences that you have illustrated. I invite the authors to expand on this issue: what is the impact of ignoring the surface scattering layer on the surface energy budget? Could the surface scattering layer be modelled? Please refer to Macfarlane et al: https://online.ucpress.edu/elementa/article/11/1/00103/195863/Evolution-of-the-microstructure-and-reflectance-of*

We followed the Reviewer's suggestion and added the following in Section 5:

"This is due to the fact that the model assumes bare ice instead of a surface scattering layer, which emerges at the top of the melting sea ice after the snow has melted. The SSL is a porous, granular, and highly fragile pillared structure on top of the ice, which effectively backscatters solar radiation and keeps the surface albedo of melting ice relatively high (Macfarlane et al. 2023). Due to the small-scale characteristics of the SSL, it is pretty difficult to relate the surface albedo of the SSL to the available variables of a climate model with spatial scales in the order of several kilometers. Consequently, the surface albedo of the SSL is a critical issue in the albedo parametrization. Since the albedo of bare ice is generally lower than the albedo of the SSL, the surplus of radiation energy at the ice surface may lead to an amplified melting of sea ice in the model."

Further, we rephrased the following (Section 3):

"We also assume that the predominantly modeled bare ice fraction with its low surface albedo contributes to the model bias. In the field, however, the surface albedo of the melting ice remained relatively high due to the presence of a brighter SSL, which is not taken into account in HIRHAM-NAOSIM."

---

## Referee Report (RR1)

Review of revised version of Jäkel et al 2023

The authors improved the paper and I very much appreciate that they diligently addressed all my comments and concerns. The language corrections were, however, insufficient. There are still grammatical and logical errors. I strongly recommend that one of the coauthors who is proficient in English will do a language check of the paper before approval for publication.

I will pick few sentences as example of badly formulated text and grammatical errors:

lines 252-253: "In general, we observed a higher albedo for cloudy conditions compared to cloudless situations, which becomes more pronounced when there is a high fraction of sea ice." This sentence is grammatically wrong: I guess "which" does not refer to "cloudless situations" as the sentence construction would imply, but to the difference in albedo between cloudy and cloudless conditions, right? Please reformulate the sentence

lines 304-306: "The smaller second modeled mode (0.77) can be attributed to grid points with a low modeled cloud coverage, such that the snow-covered ice parameter representing cloudless conditions was applied." Same problem as above: the subordinate sentence is not logically linked to the main sentence.

Moreover, the expression "such that" is used 7 times in the paper, I believe all 7 times in the wrong way. It means "to the extent that", but it seems that it is used to mean "so that", or "hence"

line 352: please replace "This is different to …" with "This differs from…"

line 389: please replace "Different to" in "Different to the radiometer measurements" with "Unlikely …" or "Differently from what showed by radiometer measurements"

lines 391-392: Please correct as "Three days after the formation of the second melt pond , …"

line 395: "albedo was significantly lower than that observed by the satellite and  measured by the RAMSES station"

line 467-472: "The maximum impact of the albedo bias on ΔFnet is derived for cloudless summer conditions (ΔFnet = ±80Wm−2). For the same range of Δα in spring, ΔFnet is found to be less than half of its magnitude in summer (ΔFnet = ±35Wm−2). In spring, however, we observed from the flight measurements an increased albedo bias with a wider distribution (Δα=0.02±0.07) than in summer (Δα=0.00±0.04). This means that greater effects on the solar radiation balance between solar incoming and outgoing irradiance due to the surface albedo model bias can be observed in summer, but these are less likely than in spring."

The last sentence refers to the larger ΔFnet in summer than in spring for the same range of Δα, and not to the immediately preceding sentence. So, I would move the sentence starting with "This means that…" before the sentence starting with "In spring…". Also, I would add few words to explain "…but these are less likely in spring": I guess you imply "as cloudless skies are rare in summer".

---

## Author Response (AR2)

**We thank the editor and the two reviewers for the time and efforts they spent in reviewing our manuscript. Please find below a discussion of the comments (italic).**
**Changes/additions made to the text are underlined and given in quotes.**

*Thank you for submitting the revised manuscript. I am pleased to tell that both reviewers are generally satisfied with your responses and have recommended this paper's publication in TC. However, Anonymous referee #2 argues that a thorough language check is still necessary before its acceptance, which I agree with. Therefore, I have judged that this paper can be published after minor revisions.*

*In addition to the review comments by Anonymous referee #2, I have checked the manuscript (v3) and listed some suggestions below. Please consider them. Before you submit a revised version of the paper, it is nice to ask coauthors again to check the paper from the standpoint of English language. Once the revised version is accepted, you are asked to attend to additional English language copy-editing by the Copernicus Publications. See more in detail at:*
*https://www.the-cryosphere.net/submission.html#english*
*I believe this paper will be more readable through the above-mentioned processes.*

Thank you for all your suggestions! We have taken them into account accordingly and made further changes to the text.

*L. 1 ~ 2: Suggest adding "in the Arctic" at the end of this sentence.*
Changed.

*L. 3~5: Too many "during" in this sentence, which makes it difficult to read. Please reformulate.*
Rephrased as follows: "The observations were conducted during five aircraft campaigns in the European Arctic at different times of the year between 2017 and 2022, one of them was part of the Multidisciplinary drifting Observatory for the Study of Arctic Climate (MOSAiC) expedition in 2020."

*L. 21: Suggest rephrasing "and therefore to" -> "and therefore forces"*
Changed.

*L. 21 ~ 22: Add "increase" after "enhances surface temperature".*
Changed.

*L. 24: "it relative impact": "its relative impact"? Also, I think "relative" can be removed.*
Changed.

*L. 36 ~ 38: "The parametrization of the albedo of the respective sea ice surface type (melt ponds, bare ice, snow) is usually based on a temperature-dependent transition between two extremes, which in the case of snow represent the albedo of dry and wet snow." The latter half of this sentence is difficult to understand. Do you mean "The albedo parametrization of the respective sea ice surface type (melt ponds, bare ice, snow) is usually based on a temperature-dependent transition between two extremes: For the case of snow, the parameterization describes a transition from the albedo of dry snow to that of wet snow."?*
Changed as follows: "The parametrization of the albedo of the respective sea ice surface types is usually based on a temperature-dependent function describing the transition between dry and wet surface conditions."

*L. 39: Suggest rephrasing "more complex surface albedo parametrizations" -> "other detailed surface albedo parametrizations"*
Changed.

*L. 62: "performed north of Svalbard" -> "performed at the north of Svalbard"*
Changed.

*L. 71: "temporal and spatial" -> "spatiotemporal"*
Changed.

*L. 102: "does not differ much" -> "does not differ so much"*
Changed.

*L. 103 ~ 104: "Station North" -> "Station Nord"?*
Changed.

*L. 106 ~ 107: "Tab. 1" -> "Table 1"*
Changed.

*L. 108: "MOSAiC-ACA. There," -> "MOSAiC-ACA, where" ?*
Changed.

*L. 112: "referred as solar" -> "referred to as solar"*
Changed.

*L. 114: "both AWI aircraft" -> "both AWI aircrafts"*
Aircraft is plural.

*L. 135: "solar surface broadband albedo" -> "surface broadband albedo"*
Changed as follows: "Since the RAMSES-ACC-VIS radiometers do not cover the entire solar spectral range, an empirical correction function was applied to convert the measured surface spectral albedo into the surface broadband albedo."

*L. 152: "Canon EOS 1D Mark III, Nikon D5" -> "Canon EOS 1D Mark III and Nikon D5"*
Changed.

*L. 156: "camera" -> "cameras"?*
Changed as follows: "which were not observed by a fisheye camera due to instrumental failures…"

*L. 228: "Temporal and spatial" -> "Spatiotemporal"*
Changed.

*L. 230: Suggest rephrasing "temporal development" -> "temporal evolution"*
Changed.

*L. 231: Suggest rephrasing "Spring time (Figs. 3a-f) was dominated by flight sections over" -> "Spring flight sections (Figs. 3a-f) were dominated by"*
Changed as follows: Flight sections in spring (Figs. 3a-f) were mostly carried out over snow or white ice (ice with a highly scattering layer on top) with surface skin temperatures below -15°C.

*L. 243: "increase" -> "increases"*
Changed as follows: "In general, the surface albedo decreases over time as a consequence of an increase of surface grain size and melt pond fraction, which are both related to the increase of skin temperature during ACLOUD (Fig. \ref{fig_frac_albedo}h)."

*L. 244: Suggest rephrasing "The first flight was dominated by sections" -> "Surface sections during the first flight were dominated by"*
Changed.

*L. 260 ~ 261: Intention is unclear. Please reformulate the sentence.*
The sentence is removed.

*Figure 3 caption: Suggest rephrasing "Temporal development of" -> "Temporal changes in"*
Changed.

*L. 294: "However, " Is it necessary?*
Removed.

*L. 295: Remove "represented by"*
Changed.

*L. 303: Suggest rephrasing "Corresponding to the maps," -> "As shown in Fig. 5b,"*
Changed as follows: "As shown in Fig. 5b, we observe greater variability in the higher resolution satellite data."

*L. 306: "As the modeled surface albedo is cloud cover dependent" -> "As the modeled surface albedo depends on cloud cover"*
Changed.

*L. 315: "as might occur" -> "as they might occur"*
Changed.

*L. 318: "With that, " -> "Therefore, "*
Changed.

*L. 323: "no significant change of the surface albedo within the time frame of the campaign were observed" -> "no significant change of the surface albedo within the time frame of the campaign were simulated"?*
Changed.

*L. 332: "Tab. 3" -> "Table 3"*
Changed.

*Figure 5 caption: Suggest rephrasing "(b) Surface albedo derived from the OLCI measurements by the MPD retrieval for 25 March 2018 (same color code as in (a))" -> "(b) Surface albedo under cloudless conditions derived from the OLCI measurements by the MPD retrieval for 25 March 2018 (same color code as in (a))"*
Changed.

*Figure 5 caption: "The aircraft measured mean surface albedo (single squares) and standard deviation (vertical bars) are given in addition." -> "The aircraft measured mean surface albedo (single squares) and standard deviation (vertical bars) are given together."*
Changed.

*Figure 5 caption: "AMSR" should be defined and introduced in Sect. 2.2*
The AMSR2 data are only used as auxiliary data (representation of the large-scale conditions). The sea ice concentration from AMSR2 has already been presented in Section 1.1 (Fig. 1).

*L. 346: "spatial and temporal" -> "spatiotemporal"*
Changed.

*L. 347: "due to" -> "attributed to"*
Changed.

*L. 351: "area-averaged measured surface albedo" -> "area-averaged surface albedo"*
Changed.

*L. 352: "standard variation" -> "standard deviation"?*
Changed.

*L. 352: "different to" -> "different from"*
Changed as follows: "This differs from …"

*L. 353: "Both, " -> "Both "*
Changed.

*L. 355 ~ 356: "significant" -> "significantly"*
Changed.

*L. 359: "play a role in" -> "affect"?*
Changed.

*L. 361: "either because of insufficient modeled snow depth or because of the relationship itself" -> "because of either insufficient modeled snow depth or the relationship itself"*
Changed.

*L. 362: Suggest rephrasing "to more deeply analyze the cause of the differences" -> "to look into the cause of the differences"*
Changed.

*L. 363: "significant" -> "significantly"*
*Changed.*

*L. 369: "in particular allow the observation of the" -> "allow to observe the"*
Changed.

*L. 373: "solar surface" -> "surface"*
Changed.

*L. 377: Suggest rephrasing "as wet snow occurred" -> "as wet snow condition prevailed"*
Changed.

*L. 387: Suggest rephrasing "which considered a higher fraction of melt ponds lowering the surface albedo" -> "which captured a higher fraction of melt ponds with lower surface albedos"*
Changed.

*L. 389: "Different to" -> "In contrast to"*
*L. 389 ~ 390: "only the transition to wet snow due to sea ice temperatures above Td leads to a drop of the surface albedo": Intention is unclear. Please reformulate.*
Changed as follows: "While the radiometer measurements showed a decrease in surface albedo due to a first melt pond event, the modeled albedo only decreased due to the transition to wet snow.

*L. 394: "covered" -> "simulated"*
Changed.

*L. 397: "surface drainage.The": Add a space between these two sentences.*
Changed.

*L. 398: "modeled (20%)" modeled what?*
Changed as follows: "The MPD OLCI satellite retrieval also determines the melt pond fraction, which was about 25 % on June 30 {Niehaus et al., 2023) and thus higher than modeled melt pond fraction (20 %)."

*L. 404: "The net solar irradiance is" -> "The net solar irradiance at the surface is"*
Changed.

*L. 412: "The correlation (R = 0.80) between the net irradiances shows a RMSE of 30.2Wm−2," -> "The correlation R between the measured and modeled net irradiances is 0.80 and RMSE of the model is 30.2 Wm−2,"*
Changed.

*L. 420: "below clouds F↓" -> "F↓ under cloudy conditions"*
Changed.

*L. 441 ~ 442: Suggest rephrasing "allows a reduction in the dependencies" -> "allows to consider reduced dependencies"*
Changed.

*L. 445: "Radiative transfer simulations" -> "Radiative transfer simulations (Appendix A)"*
Changed.

*L. 447: "were matched to" -> "matched"*
Changed.

*L. 459: "observed" -> "obtained"*

*Changed.*

*L. 460: "compared to the HIRHAM-NAOSIM run": Can be removed.*
Changed.

*L. 464: Suggest rephrasing "a surface albedo that is dominated by wet snow" -> "low surface albedo due to wet snow"*
Changed.

*L. 470 ~ 472: Suggest rephrasing "This means that greater effects on the solar radiation balance between solar incoming and outgoing irradiance due to the surface albedo model bias can be observed in summer, but these are less likely than in spring." -> "This means that the bias of modeled surface albedo can have greater effects on the simulated net solar irradiance at the surface in summer compared to spring."*
Changed.

*L. 482 ~ 483: Suggest adding "applied in the Arctic" at the end of this sentence.*
Changed.

*L. 493 ~ 494: "values between 0.8 and 0.9 are reported" -> "ranges between 0.8 and 0.9"*
Changed.

*L. 496: "in cases of clouds" -> "under cloudy conditions"*
Changed.

*L. 504: "this approach, uses COD": Comma should be removed.*
Changed.

*L. 510 ~ 511: Suggest rephrasing "we found that it was primarily the uncertainties in the parametrization of the surface types that affected the outcome of the modeled surface albedo" -> "we found that the error of modeled surface albedo was primarily attributed to the uncertainties in the surface type parametrization"*
Changed.

*L. 514: Suggest rephrasing "Larger surface albedo differences (more than 0.1) were observed after the drainage of the observed melt ponds end of June." -> "Larger surface albedo differences (more than 0.1) were obtained after the drainage of the observed melt ponds at the end of June."*
Changed.

*L. 518: "surface scattering layer" -> "surface scattering layer (SSL)"*
Changed.

*L. 521: Remove "pretty", as it sounds subjective*
Changed.

*L. 529: "less well observed" -> "less captured"*
Changed.

*L. 531: Remove "quite", as it sounds subjective.*

Changed.

*L. 535 ~ 536: "This we primarily attribute to differences in surface albedo, but also partly to uncertainties in the modeled cloud cover as derived from the ranking of the standardized regression coefficients.": Difficult to understand. Please reformulate.*

Changed as follows: "This error can be primarily attributed to differences in surface albedo. However, the ranking of the standardized regression coefficient suggests coefficient suggests that uncertainties in the modeled cloud cover also contribute to the model bias in net irradiance."

*Review of revised version of Jäkel et al 2023*

*The authors improved the paper and I very much appreciate that they diligently addressed all my comments and concerns. The language corrections were, however, insufficient. There are still grammatical and logical errors. I strongly recommend that one of the coauthors who is proficient in English will do a language check of the paper before approval for publication.*

*I will pick few sentences as example of badly formulated text and grammatical errors:*

*lines 252-253: "In general, we observed a higher albedo for cloudy conditions compared to cloudless situations, which becomes more pronounced when there is a high fraction of sea ice." This sentence is grammatically wrong: I guess "which" does not refer to "cloudless situations" as the sentence construction would imply, but to the difference in albedo between cloudy and cloudless conditions, right? Please reformulate the sentence*
We changed it as follows: "In general, we observed a higher albedo for the same amount of sea ice under cloudy conditions than under cloudless conditions. This effect was more pronounced when a high proportion of sea ice was present."

*lines 304-306: "The smaller second modeled mode (0.77) can be attributed to grid points with a low modeled cloud coverage, such that the snow-covered ice parameter representing cloudless conditions was applied." Same problem as above: the subordinate sentence is not logically linked to the main sentence. Moreover, the expression "such that" is used 7 times in the paper, I believe all 7 times in the wrong way. It means "to the extent that", but it seems that it is used to mean "so that", or "hence"*
Changed from "such that" → "hence". "such that" has also been replaced everywhere else.

*line 352: please replace "This is different to …" with "This differs from…"*
Changed.

*line 389: please replace "Different to" in "Different to the radiometer measurements" with "Unlikely …" or "Differently from what showed by radiometer measurements"*
Changed as follows: "While the radiometer measurements showed a decrease in surface albedo due to a first melt pond event, the modeled albedo only decreased due to the transition to wet snow."

*lines 391-392: Please correct as "Three days after the formation of the second melt pond in the field measurements, …"*
Changed as follows: "Three days after the observed formation of the second melt pond, …"

*line 395: "albedo was significantly lower than that observed by the satellite and even more lower than that measured by the RAMSES station"*

Changed as follows: "After the formation of melt ponds, however, the modeled surface albedo was significantly underestimated compared to the observations by the satellite and the ground-based RAMSES station."

*line 467-472: "The maximum impact of the albedo bias on ΔFnet is derived for cloudless summer conditions (ΔFnet = ±80Wm−2). For the same range of Δα in spring, ΔFnet is found to be less than half of its magnitude in summer (ΔFnet = ±35Wm−2). In spring, however, we observed from the flight measurements an increased albedo bias with a wider distribution (Δα=0.02±0.07) than in summer (Δα=0.00±0.04). This means that greater effects on the solar radiation balance between solar incoming and outgoing irradiance due to the surface albedo model bias can be observed in summer, but these are less likely than in spring."*

*The last sentence refers to the larger ΔFnet in summer than in spring for the same range of Δα, and not to the immediately preceding sentence. So, I would move the sentence starting with "This means*

*that…" before the sentence starting with "In spring…". Also, I would add few words to explain "…but these are less likely in spring": I guess you imply "as cloudless skies are rare in summer".*

Changed as follows: "For the same range of Δα in spring, ΔFnet is found to be less than half of its magnitude in summer (ΔFnet = ±35Wm−2). This means that the bias of modeled surface albedo can have greater effects on the simulated net solar irradiance at the surface in summer compared to spring. In spring, however, we observed from the flight measurements an increased albedo bias with a wider distribution (Δα=0.02±0.07) than in summer (Δα=0.00±0.04). The deviation from…"